# How Much Pre-training Is Enough to Discover a Good Subnetwork?

**Cameron R. Wolfe**[*]                                          *crw13@rice.edu*
*Department of Computer Science*
*Rice University*

**Fangshuo Liao**[*†]                                          *Fangshuo.Liao@rice.edu*
*Department of Computer Science*
*Rice University*

**Qihan Wang**                                          *Chuck.Wang@rice.edu*
*Department of Computer Science*
*Rice University*

**Junhyung Lyle Kim**                                          *jlylekim@rice.edu*
*Department of Computer Science*
*Rice University*

**Anastasios Kyrillidis**                                          *anastasios@rice.edu*
*Department of Computer Science*
*Rice University*

**Reviewed on OpenReview:** *https://openreview.net/forum?id=UVE7LllpXe*

## Abstract

Neural network pruning helps discover efficient, high-performing subnetworks within pre-trained, dense network architectures. More often than not, it involves a three-step process—pre-training, pruning, and re-training—that is computationally expensive, as the dense model must be fully pre-trained. While previous work has revealed through experiments the relationship between the amount of pre-training and the performance of the pruned network, a theoretical characterization of such dependency is still missing. Aiming to mathematically analyze the amount of dense network pre-training needed for a pruned network to perform well, we discover a simple theoretical bound in the number of gradient descent pre-training iterations on a two-layer fully connected network in the NTK regime, beyond which pruning via greedy forward selection (Ye et al., 2020) yields a subnetwork that achieves good training error. Interestingly, this threshold is logarithmically dependent upon the size of the dataset, meaning that experiments with larger datasets require more pre-training for subnetworks obtained via pruning to perform well. Lastly, we empirically validate our theoretical results on multi-layer perceptions and residual-based convolutional networks trained on MNIST, CIFAR, and ImageNet datasets.

## 1 Introduction

Neural network pruning refers to dropping weights in a large neural network without significantly degrading the model's performance. It has been widely applied to model compression (Anwar et al., 2015; Liang et al., 2021; Li et al., 2023; Blalock et al., 2020). Neural network pruning usually involves three steps: *i) pre-training*, where a large, randomly initialized neural network is trained on a dataset to reach a certain test

---

[*]Equal Contribution
[†]Corresponding Author

accuracy; *ii) pruning*, where a subset of the neural network weights are dropped; and *iii) re-training*, where the pruned neural network is trained again to maintain the desired accuracy. While some methods perform only a subset of these steps, most existing algorithms include at least the pre-training and pruning phases.

The above three-step procedure mainly originates from the prominent line of work of the Lottery Ticket Hypothesis (Frankle & Carbin, 2018; Chen et al., 2020; Frankle et al., 2019; Gale et al., 2019; Liu et al., 2018; Morcos et al., 2019; Zhou et al., 2019; Zhu & Gupta, 2017) (or LTH): i.e., the idea that a pre-trained model contains "lottery tickets" (i.e., smaller subnetworks) such that if we select those "tickets" cleverly, those submodels do not lose much in accuracy while reducing significantly the size of the model. To circumvent the cost of pre-training, several works explore the possibility of pruning networks directly from initialization (i.e., the "strong lottery ticket hypothesis") (Frankle et al., 2020; Ramanujan et al., 2019; Pensia et al., 2021; Xiong et al., 2022), but subnetwork performance could suffer. Adopting a hybrid approach, good subnetworks can also be obtained from models with minimal pre-training (Chen et al., 2020; You et al., 2019) (i.e., "early-bird" tickets): i.e., the pre-training step, though indispensable, needs to be executed for only a small extent before the pruning step can find a small model that performs well (namely the winning ticket). This line of work further promoted the application of pruning algorithms to large neural networks. However, despite its strong support from empirical observation, the relationship between pre-training and pruning has never been examined theoretically, even for simple neural architectures.

This paper aims to fill this gap by bridging the theory of neural networks trained with gradient descent and the greedy forward selection pruning algorithm. Our analysis focuses on a two-layer neural network with initialization and over-parameterization in the NTK regime. From this analysis, *we discover a simple threshold in the number of pre-training iterations—logarithmically dependent upon the size of the dataset—beyond which subnetworks obtained via greedy forward selection perform well in terms of training error*. Such a finding offers a theoretical insight into the early-bird ticket phenomenon and provides intuition for why discovering high-performing subnetworks is more difficult in large-scale experiments (You et al., 2019; Renda et al., 2020; Liu et al., 2018). In particular, our contributions can be summarized as below:

- Based on the analysis of Ye et al. (2020), our Lemma 1 serves as an improvement over the proof in Ye et al. (2020): Our proof shows that the greedy forward selection error depends on both the initial selection error, the maximum distance of neuron separation, and the loss of the current dense neural network at pruning time. In particular, the greedy forward selection error has a $O\left(\frac{\log k}{k}\right)$ dependency on the first two terms, where $k$ is the number of neurons selected.

- By analyzing the neural network weight change during training, we find out that the initial selection error and the maximum distance of neuron separation can be bounded by $O\left(\sqrt{N}\right)$ throughout training. This implies that the effect of the two terms can be mitigated as long as the number of selected neurons satisfies $k = \Omega\left(\sqrt{N}\log N\right)$.

- Lastly, we connect the training loss of the dense neural network at pruning time with the existing theory that characterizes the convergence of neural network training loss. We derive that, to choose $k$ neurons in the greedy forward selection, it is necessary to perform at least $O\left(\log k\right)$ pre-training steps to guarantee improvement during the greedy forward selection. Therefore, our result identifies the connection between the number of pre-training steps required and the target loss after pruning.

## 2 Background and Problem Setup

**Notation.** Vectors are represented with bold type (e.g., $\mathbf{x}$), while scalars are represented by normal type (e.g., $x$). $\|\cdot\|_2$ represents the $\ell_2$ vector norm. For a function $\phi : \mathbb{R} \to \mathbb{R}$, we use $\|\phi\|_{\mathcal{H}}$ to represent the Hermite norm of $\phi$ (see Definition 4 of Song et al. (2021)). $[N]$ is used to represent the set of positive integers from 1 to $N$ (i.e., $[N] = \{1 \ldots N\}$).

**Network Parameterization.** We consider a two-layer neural network with $N$ hidden neurons for simplicity. In particular, given an input vector $\mathbf{x} \in \mathbb{R}^d$, the activation of the $i$-th hidden neuron, $\sigma\left(\cdot, \boldsymbol{\theta}_i\right)$, is given by:

$$\sigma\left(\mathbf{x}, \boldsymbol{\theta}_i\right) = N \cdot b_i \sigma_+ \left(\mathbf{a}_i^\top \mathbf{x}\right). \tag{1}$$

Here, $\sigma_+$ is the activation function. The weights associated with the $i$-th neuron are concatenated into $\boldsymbol{\theta}_i = [b_i, \mathbf{a}_i]$, where $\mathbf{a}_i \in \mathbb{R}^d$ is the first layer weight, and $b_i \in \mathbb{R}$ is the second layer weight. Given input $\mathbf{x}$, the neural network output $f(\mathbf{x}, \boldsymbol{\Theta})$ can be viewed as the average of the hidden neuron's activation:

$$f(\mathbf{x}, \boldsymbol{\Theta}) = \frac{1}{N} \sum_{i=1}^{N} \sigma(\mathbf{x}, \boldsymbol{\theta}_i), \tag{2}$$

where $\boldsymbol{\Theta} = \{\boldsymbol{\theta}_1, \ldots, \boldsymbol{\theta}_N\}$ represents all weights within the two-layer neural network. Two-layer neural networks with the form in (2) have been studied extensively as a simple yet representative instance of deep learning models (Du et al., 2019; Oymak & Soltanolkotabi, 2019; Song et al., 2021).

In this paper, we shall consider the scheme of neuron pruning. In particular, a pruned network is defined by a subset of the hidden neurons $\mathcal{S} \subseteq [N]$ as:

$$f_\mathcal{S}(\mathbf{x}, \boldsymbol{\Theta}) = \frac{1}{|\mathcal{S}|} \sum_{i \in \mathcal{S}} \sigma(\mathbf{x}, \boldsymbol{\theta}_i). \tag{3}$$

As a special case, we note that the whole network can also be written as $f(\mathbf{x}, \boldsymbol{\Theta}) = f_{[N]}(\mathbf{x}, \boldsymbol{\Theta})$. We make the following assumption about the neural network:

**Assumption 1.** *(Neural Network) There exist $\delta > 0$ and some $r_1, r_2 \in \mathbb{R}$, such that $\sigma_+(0) = 0, |\sigma_+(\cdot)| \leq 1, |\sigma'_+(\cdot)| \leq \delta$ and $|\sigma''_+(\cdot)| \leq \delta$, $\tau^{r_1} |\sigma_+(a)| \leq |\sigma_+(\tau a)| \leq \tau^{r_2} |\sigma_+(a)|$ for all $a \in \mathbb{R}$ and $\tau \in (0,1)$, and $\|\sigma_+\|_\mathcal{H} < \infty$ for $\sigma_+$ defined in (1). Moreover, before pre-training, the weights of the neural network are initialized according to $b_i \sim \mathcal{N}(0, \omega_b^2)$ and $\mathbf{a}_i \sim \mathcal{N}(0, \omega_a^2 \mathbf{I}_d)$, for some $\omega_a, \omega_b > 0$.*

Compared with the assumption on the activation function in (Song et al., 2021), we added the additional assumption $|\sigma_+(\cdot)| \leq 1$. An example of the activation function that satisfies Assumption 1 is the $\tanh(\cdot)$ function. While ReLU is also used widely in practice, in this paper we focus on the family of bounded smooth activation functions for the simplicity of the analysis, as in previous works Liu et al. (2021); Song et al. (2021); Liu et al. (2023).

**Dataset.** We assume that our network is modeling a dataset $D = \{(\mathbf{x}_j, y_j)\}_{j=1}^m$ with $m$ input-output pairs, where $\mathbf{x}_j \in \mathbb{R}^d$ and $y_j \in \mathbb{R}$ for each $j \in [m]$ satisfy the following assumption:

**Assumption 2.** *(Data) The input and label of the dataset are bounded as $\|\mathbf{x}_j\|_2 \leq 1$ for all $j \in [m]$ and $\sum_{j=1}^m y_j^2 \leq 1$.*

For a given network $f_\mathcal{S}(\mathbf{x}, \boldsymbol{\Theta})$, we consider the $\ell_2$-norm regression loss over the dataset:

$$\mathcal{L}[f_\mathcal{S}(\cdot, \boldsymbol{\Theta})] = \frac{1}{2} \sum_{j=1}^m (f_\mathcal{S}(\mathbf{x}_j, \boldsymbol{\Theta}) - y_j)^2. \tag{4}$$

At pre-training, we use gradient descent (GD) over $\boldsymbol{\Theta}$ to minimize the whole network loss $\mathcal{L}[f(\cdot, \boldsymbol{\Theta})]$:

$$\boldsymbol{\Theta}_{t+1} = \boldsymbol{\Theta}_t - \eta \nabla_{\boldsymbol{\Theta}} \mathcal{L}[f(\cdot, \boldsymbol{\Theta}_t)]. \tag{5}$$

During pruning, we will track how the subnetwork loss $\mathcal{L}[f_\mathcal{S}(\cdot, \boldsymbol{\Theta})]$ changes as we change the hidden neuron subset $\mathcal{S}$. We will discuss this in more detail in the section below.

## 3 Pruning with Greedy Forward Selection

In this section, we focus on a specific and straightforward algorithm adopted in the pruning stage, namely the Greedy Forward Selection proposed by Ye et al. (2020) (see Algorithm 1). Since the neural network weights are fixed in the pruning stage, our discussion in this section will assume a set of given weights $\boldsymbol{\Theta}$. Starting from an empty subnetwork (i.e., $\mathcal{S} = \emptyset$), we aim to discover a subset of neurons $\mathcal{S}^\star$ given by:

$$\mathcal{S}^\star = \arg\min_{\mathcal{S} \subseteq [N]} \mathcal{L}[f_\mathcal{S}(\cdot, \boldsymbol{\Theta})]; \quad |\mathcal{S}^\star| \ll N. \tag{6}$$

Instead of discovering an exact solution to this challenging combinatorial optimization problem, Algorithm 1 is used to find an approximate solution. At each iteration $k$, we select the neuron that yields the most significant decrease in loss. Since Algorithm 1 is an approximation to the optimal solution by its nature, instead of focusing on its optimality, we will investigate the property of $\mathcal{L}[f_{\mathcal{S}}(\cdot, \boldsymbol{\Theta})]$ when $\mathcal{S}$ is returned by Algorithm 1.

Similar to Ye et al. (2020), we shall consider the following interpretation from a geometric perspective to provide an analysis of Algorithm 1. We define $\mathbf{y} = [y_1, y_2, \ldots, y_m]$, representing a concatenated vector of all labels within the dataset. Similarly, we define $\phi_{i,j} = \sigma(\mathbf{x}_j, \boldsymbol{\theta}_i)$ as the output of neuron $i$ for the $j$-th input vector in the dataset and construct the vector $\boldsymbol{\Phi}_i = [\phi_{i,1}, \phi_{i,2}, \ldots, \phi_{i,m}]$, which is a concatenated vector of output activations for a single neuron across the entire dataset. The outputs of a pruned network $\hat{\mathbf{y}} = [f_{\mathcal{S}}(\mathbf{x}_1, \boldsymbol{\Theta}), \ldots, f_{\mathcal{S}}(\mathbf{x}_m, \boldsymbol{\Theta})]$ can then be viewed as a convex combination of $\boldsymbol{\Phi}_1, \ldots, \boldsymbol{\Phi}_N$. We

---

**Algorithm 1** Greedy Forward Selection

1: $\mathcal{S}_0 := \emptyset$
2: **for** $k = 1, 2, \ldots$ **do**
3:      # Select a new neuron
4:      $i_k := \arg\min_{i \in [N]} \mathcal{L}\left[f_{\mathcal{S}_{k-1} \cup \{i\}}(\cdot, \boldsymbol{\Theta})\right]$
5:      # Add neuron to the subnetwork
6:      $\mathcal{S}_k := \mathcal{S}_{k-1} \cup \{i_k\}$
7: **end for**
8: **return** $\mathcal{S}$

---

use $\mathcal{M}_N$ to denote the convex hull over such activation vectors for all $N$ neurons, and, with a slight abuse of notation, use $\texttt{Vert}(\mathcal{M}_N)$ to denote the set of neuron outputs $\boldsymbol{\Phi}_1, \ldots, \boldsymbol{\Phi}_m$. Notice that $\texttt{Vert}(\mathcal{M}_N)$ must cover the vertices of $\mathcal{M}_N$:

$$\mathcal{M}_N = \texttt{Conv}\left\{\boldsymbol{\Phi}_i : i \in [N]\right\}; \quad \texttt{Vert}(\mathcal{M}_N) = \left\{\boldsymbol{\Phi}_i : i \in [N]\right\}. \tag{7}$$

Intuitively, $\mathcal{M}_N$ forms a marginal polytope of the feature map for all neurons in the two-layer network across every data point. Using the construction $\mathcal{M}_N$, the $\ell_2$ loss can be written as follows:

$$\ell(\mathbf{z}) = \frac{1}{2}\|\mathbf{z} - \mathbf{y}\|^2; \quad \mathbf{z} \in \mathcal{M}_N. \tag{8}$$

With this geometric interpretation of the neural network output, we can relax the combinatorial problem in (6) to $\min_{\mathbf{z} \in \mathcal{M}_N} \ell(\mathbf{z})$. Moreover, using this construction, we can write the update rule for Algorithm 1 as:

$$\text{(Select new neuron):} \quad \mathbf{q}_k = \arg\min_{\mathbf{q} \in \texttt{Vert}(\mathcal{M}_n)} \ell\left(\frac{1}{k} \cdot (\mathbf{z}_{k-1} + \mathbf{q})\right) \tag{9}$$

$$\text{(Add neuron to subnetwork):} \quad \mathbf{z}_k = \mathbf{z}_{k-1} + \mathbf{q}_k \tag{10}$$

$$\text{(Uniform average of neuron outputs):} \quad \mathbf{u}_k = \frac{1}{k} \cdot \mathbf{z}_k. \tag{11}$$

In words, (9)-(11) include the output of a new neuron, given by $\mathbf{q}_k$, within the current subnetwork at each pruning iteration based on a greedy minimization of the loss $\ell(\cdot)$. Then, the output of the pruned subnetwork over the dataset at the $k$-th iteration, given by $\mathbf{u}_k$, is computed by taking a uniform average over the activation vectors of the $k$ active neurons in $\mathbf{z}_k$. From this perspective, we have that $\mathbf{u}_k = [f_{\mathcal{S}_k}(\mathbf{x}_1, \boldsymbol{\Theta}), \ldots, f_{\mathcal{S}_k}(\mathbf{x}_m, \boldsymbol{\Theta})]$. Notably, the procedure in (9)-(11) can select the same neuron multiple times during successive pruning iterations. Such selection with replacement can be interpreted as a form of training during pruning—multiple selections of the same neuron are equivalent to modifying the neuron's output layer weight $b_i$ in (1). Nonetheless, we highlight that such "training" does not violate the core purpose of pruning: *we still obtain a smaller subnetwork with performance comparable to the dense network from which it was derived.*

## 4 How Much Pre-training Do We Really Need?

As previously stated, no existing theoretical analysis has quantified the impact of pre-training on the performance of a pruned subnetwork. Here, we consider this problem by extending analysis for pruning via greedy forward selection to determine the relationship between GD pre-training and subnetwork training loss. For the convenience of our analysis, for a fixed set of neural network weights $\boldsymbol{\Theta}$, we define $\mathcal{D}_{\mathcal{M}_N} = \max_{\mathbf{u}, \mathbf{v} \in \mathcal{M}_N} \|\mathbf{u} - \mathbf{v}\|_2$. Our first lemma characterizes the training loss convergence during the pruning phase based on a general neural network state.

**Lemma 1.** *Fix the weights $\boldsymbol{\Theta}$ of the two-layer neural network $f(\cdot, \boldsymbol{\Theta})$ defined in (2). Then, Algorithm 1 generates the sequence $\{\mathcal{S}_k\}_{k=1}^{\infty}$ satisfying*

$$\mathcal{L}\left[f_{\mathcal{S}_k}\left(\cdot, \boldsymbol{\Theta}\right)\right] \le \frac{1}{k}\mathcal{L}\left[f_{\mathcal{S}_1}\left(\cdot, \boldsymbol{\Theta}\right)\right] + \frac{1 + \log k}{2k}\mathcal{D}_{\mathcal{M}_N}^2 + \frac{k-1}{k}\mathcal{L}\left[f\left(\cdot, \boldsymbol{\Theta}\right)\right]. \tag{12}$$

*Proof.* The idea is similar to Ye et al. (2020). To start, we define $\mathbf{q}_k'$ and $\mathbf{u}_k'$ as follows:

$$\mathbf{q}_k' = \underset{\mathbf{u} \in \mathcal{M}_N}{\arg\min} \left\langle \nabla\ell\left(\mathbf{u}_{k-1}\right), \mathbf{u}\right\rangle; \quad \mathbf{u}_k' = \frac{1}{k}\left(\mathbf{z}_{k-1} + \mathbf{q}_k'\right).$$

Since $\mathbf{q}_k'$ is the minimizer of a linear objective in a polytope, we must have that $\mathbf{q}_k' \in \texttt{Vert}\left(\mathcal{M}_N\right)$. Therefore, by (9) and (10), and due to the optimality of $\mathbf{q}_k$, we must have that:

$$\ell\left(\mathbf{u}_k\right) = \ell\left(\tfrac{1}{k}\left(\mathbf{z}_{k-1} + \mathbf{q}_k\right)\right) \le \ell\left(\tfrac{1}{k}\left(\mathbf{z}_{k-1} + \mathbf{q}_k'\right)\right) = \ell\left(\mathbf{u}_k'\right).$$

We further notice that the objective in (8) is quadratic. Therefore, we have:

$$\ell\left(\mathbf{u}_k\right) \le \ell\left(\mathbf{u}_k'\right) = \ell\left(\mathbf{u}_{k-1}\right) + \left\langle\nabla\ell\left(\mathbf{u}_{k-1}\right), \mathbf{u}_k' - \mathbf{u}_{k-1}\right\rangle + \frac{1}{2}\left\|\mathbf{u}_k' - \mathbf{u}_{k-1}\right\|_2^2. \tag{13}$$

Moreover, (11) implies $\mathbf{u}_k' - \mathbf{u}_{k-1} = \frac{k-1}{k}\mathbf{u}_{k-1} + \frac{1}{k}\mathbf{q}_k' - \mathbf{u}_{k-1} = \frac{1}{k}\left(\mathbf{q}_k' - \mathbf{u}_{k-1}\right)$. Therefore (13) becomes:

$$\ell\left(\mathbf{u}_k\right) \le \ell\left(\mathbf{u}_{k-1}\right) + \frac{1}{k}\left\langle\nabla\ell\left(\mathbf{u}_{k-1}\right), \mathbf{q}_k' - \mathbf{u}_{k-1}\right\rangle + \frac{1}{2k^2}\left\|\mathbf{q}_k' - \mathbf{u}_{k-1}\right\|_2^2. \tag{14}$$

Again, since the objective in (8) is quadratic, we must have that $\ell$ is convex. Therefore:

$$\min_{\mathbf{u} \in \mathcal{M}_N} \ell\left(\mathbf{u}\right) \ge \min_{\mathbf{u} \in \mathcal{M}_N} \left\{\ell\left(\mathbf{u}_{k-1}\right) + \left\langle\nabla\ell\left(\mathbf{u}_{k-1}\right), \mathbf{u} - \mathbf{u}_{k-1}\right\rangle\right\}$$

$$\ge \ell\left(\mathbf{u}_{k-1}\right) + \left\langle\nabla\ell\left(\mathbf{u}_{k-1}\right), \mathbf{q}_k' - \mathbf{u}_{k-1}\right\rangle.$$

Since $\hat{\mathbf{y}} = [f(\mathbf{x}_1, \boldsymbol{\Theta}), \dots, f(\mathbf{x}_m, \boldsymbol{\Theta})] = \frac{1}{N}\sum_{i=1}^N \boldsymbol{\Phi}_i \in \mathcal{M}_N$, we must have that:

$$\mathcal{L}\left[f\left(\cdot, \boldsymbol{\Theta}\right)\right] = \ell\left(\hat{\mathbf{y}}\right) \ge \min_{\mathbf{u} \in \mathcal{M}_N} \ell\left(\mathbf{u}\right) \ge \ell\left(\mathbf{u}_{k-1}\right) + \left\langle\nabla\ell\left(\mathbf{u}_{k-1}\right), \mathbf{q}_k' - \mathbf{u}_{k-1}\right\rangle. \tag{15}$$

Plugging (15) into (14) and noticing that $\left\|\mathbf{q}_k' - \mathbf{u}_{k-1}\right\|_2 \le \mathcal{D}_{\mathcal{M}_N}$ gives:

$$\ell\left(\mathbf{u}_k\right) - \mathcal{L}\left[f\left(\cdot, \boldsymbol{\Theta}\right)\right] \le \left(1 - \frac{1}{k}\right)\ell\left(\mathbf{u}_{k-1}\right) + \frac{1}{k}\mathcal{L}\left[f\left(\cdot, \boldsymbol{\Theta}\right)\right] + \frac{1}{2k^2}\mathcal{D}_{\mathcal{M}_N}^2 - \mathcal{L}\left[f\left(\cdot, \boldsymbol{\Theta}\right)\right]$$

$$\le \left(1 - \frac{1}{k}\right)\left(\ell\left(\mathbf{u}_{k-1}\right) - \mathcal{L}\left[f\left(\cdot, \boldsymbol{\Theta}\right)\right]\right) + \frac{1}{2k^2}\mathcal{D}_{\mathcal{M}_N}^2$$

Unrolling the iterates gives:

$$\ell\left(\mathbf{u}_k\right) - \mathcal{L}\left[f\left(\cdot, \boldsymbol{\Theta}\right)\right] \le \prod_{t=1}^k \left(1 - \frac{1}{k}\right)\left(\ell\left(\mathbf{u}_1\right) - \mathcal{L}\left[f\left(\cdot, \boldsymbol{\Theta}\right)\right]\right) + \frac{\mathcal{D}_{\mathcal{M}_N}^2}{2}\sum_{t=1}^k t^{-2}\prod_{j=t+1}^k\left(1 - \frac{1}{j}\right)$$

$$= \prod_{t=1}^k \frac{k-1}{k}\left(\ell\left(\mathbf{u}_1\right) - \mathcal{L}\left[f\left(\cdot, \boldsymbol{\Theta}\right)\right]\right) + \frac{\mathcal{D}_{\mathcal{M}_N}^2}{2}\sum_{t=1}^k t^{-2}\prod_{j=t+1}^k\frac{j-1}{j}$$

$$= \frac{1}{k}\left(\ell\left(\mathbf{u}_1\right) - \mathcal{L}\left[f\left(\cdot, \boldsymbol{\Theta}\right)\right]\right) + \frac{\mathcal{D}_{\mathcal{M}_N}^2}{2k}\sum_{t=1}^k t^{-1}$$

$$\le \frac{1}{k}\left(\ell\left(\mathbf{u}_1\right) - \mathcal{L}\left[f\left(\cdot, \boldsymbol{\Theta}\right)\right]\right) + \frac{1 + \log k}{2k}\mathcal{D}_{\mathcal{M}_N}^2$$

This shows that:

$$\ell\left(\mathbf{u}_k\right) \le \frac{1}{k}\ell\left(\mathbf{u}_1\right) + \frac{1 + \log k}{2k}\mathcal{D}_{\mathcal{M}_N}^2 + \frac{k-1}{k}\mathcal{L}\left[f\left(\cdot, \boldsymbol{\Theta}\right)\right].$$

Plugging in $\mathcal{L}\left[f_{\mathcal{S}_k}\left(\cdot, \boldsymbol{\Theta}\right)\right] = \ell\left(\mathbf{u}_k\right)$ and $\mathcal{L}\left[f_{\mathcal{S}_1}\left(\cdot, \boldsymbol{\Theta}\right)\right] = \ell\left(\mathbf{u}_1\right)$ gives the desired result. $\qquad\square$

Lemma 1 characterizes the pruned network loss using a combination of three terms. Notably, the first and second terms decrease as the number of greedy forward selection steps $k$ grows. In the limit of $k \to \infty$, the pruned network loss $\mathcal{L}\left[f_{\mathcal{S}_k}\left(\cdot, \boldsymbol{\Theta}\right)\right]$ decreases to the loss achieved by the dense network $\mathcal{L}\left[f\left(\cdot, \boldsymbol{\Theta}\right)\right]$. However, to guarantee the sparsity of the pruned network, we need the first and second terms in (12) to decrease to a meaningfully small scale within a moderate number of greedy forward selection steps. This requires us to provide an upper bound of both $\mathcal{L}\left[f_{\mathcal{S}_1}\left(\cdot, \boldsymbol{\Theta}\right)\right]$ and $\mathcal{D}_{\mathcal{M}_N}$.

**Remark.** Although our proof takes a similar strategy as in Ye et al. (2020), it should be noted that the proof in Ye et al. (2020) has an issue that leads to an incorrect result.[1] Our proof fixes this issue, which leads to a seemingly worse bound compared with Ye et al. (2020).

### 4.1 Bounding Initial Pruning Loss and the Diameter

In this subsection, we shall focus on the upper bound of $\mathcal{L}\left[f_{\mathcal{S}_1}\left(\cdot, \boldsymbol{\Theta}\right)\right]$ and $\mathcal{D}_{\mathcal{M}_N}$. Recall that we define $\mathcal{D}_{\mathcal{M}_N} = \max_{\mathbf{u}, \mathbf{v} \in \mathcal{M}_N} \|\mathbf{u} - \mathbf{v}\|_2$. Since $\mathcal{M}_N$ is defined as the convex hull over $\boldsymbol{\Phi}_1, \ldots, \boldsymbol{\Phi}_N$, we can show the $\mathcal{D}_{\mathcal{M}_N}$ can be controlled by the maximum difference between two vertices. We formulate this idea in the lemma below.

**Lemma 2.** *Let $\mathcal{D}' = \max_{i,j \in [N]} \|\boldsymbol{\Phi}_i - \boldsymbol{\Phi}_j\|_2$. Then we have $\mathcal{D}' \geq \mathcal{D}_{\mathcal{M}_N}$.*

Since this is a standard result for convex polytope, we defer the proof to Appendix A. By Lemma 2, to provide an upper bound on $\mathcal{D}_{\mathcal{M}_N}$, it suffice to bound $\mathcal{D}'$. Applying the triangle inequality, we can upper bound $\mathcal{D}'$ by

$$\mathcal{D}' \leq \max_{i,j \in [N]} \left(\|\boldsymbol{\Phi}_i\|_2 + \|\boldsymbol{\Phi}_j\|_2\right) \leq 2 \max_{i \in [N]} \|\boldsymbol{\Phi}_i\|_2$$

Recall that $\boldsymbol{\Phi}_i$ is the hidden neuron output over all input vectors. Therefore, $\|\boldsymbol{\Phi}_i\|_2$ depends on the weight of the neural network. However, to further incorporate the loss dynamic of the pre-training into the analysis, we have to develop a universal upper bound on $\|\boldsymbol{\Phi}_i\|_2$ for all weights in the trajectory of the gradient descent. With sufficient over-parameterization, we can bound $\|\boldsymbol{\Phi}_i\|_2$ at any time step of the gradient descent pre-training.

**Lemma 3.** *Let Assumption 1 and Assumption 2 hold with $\omega_b \leq O\left(m^{\frac{1}{2}} N^{-\frac{3}{4}}\right)$. Let $\boldsymbol{\Phi}_1, \ldots, \boldsymbol{\Phi}_N$ be the hidden neuron outputs defined over any neural network weights $\boldsymbol{\Theta} = \boldsymbol{\Theta}_t$ for $t = 1, 2, \ldots$. If the number of neurons $N = \Omega\left(\frac{m^4}{\lambda_{\min}^8} \mathcal{L}\left[f\left(\cdot, \boldsymbol{\Theta}_0\right)\right]^2\right)$, then with high probability we have that $\|\boldsymbol{\Phi}_i\|_2 \leq C_3 N^{\frac{1}{4}}$ for some constant $C_3 > 0$.*

*Proof.* Let $\boldsymbol{\Phi}_i$ denote the outputs of neuron $i$ for the whole dataset at an arbitrary time step $t$ during the pre-training. That is, for an arbitrary $t$, we consider $\boldsymbol{\Phi}_i = [\sigma\left(\mathbf{x}_1, \boldsymbol{\theta}_{i,t}\right), \ldots, \sigma\left(\mathbf{x}_m, \boldsymbol{\theta}_{i,t}\right)]$. Then we have that:

$$\|\boldsymbol{\Phi}_i\|_2 = \left(\sum_{j=1}^{m} \sigma\left(\mathbf{x}_j, \boldsymbol{\theta}_{i,t}\right)^2\right)^{\frac{1}{2}} \leq \sqrt{m} \max_{j \in [m]} |\sigma\left(\mathbf{x}_j, \boldsymbol{\theta}_{i,t}\right)|. \tag{16}$$

Recall that $\sigma\left(\mathbf{x}_j, \boldsymbol{\theta}_{i,t}\right) = N b_{i,t} \sigma_+\left(\mathbf{a}_{i,t}^\top \mathbf{x}_j\right)$. By our choice of $\sigma_+$ in Assumption 1, we must have that $\left|\sigma_+\left(\mathbf{a}_{i,t}^\top \mathbf{x}_j\right)\right| \leq 1$. Therefore, (16) becomes:

$$\|\boldsymbol{\Phi}_i\|_2 \leq \sqrt{m} N |b_{i,t}|. \tag{17}$$

It boils down to bounding $|b_{i,t}|$. We use the following decomposition:

$$|b_{i,t}| \leq |b_{i,0}| + |b_{i,t} - b_{i,0}|.$$

Since $b_{i,0} \sim \mathcal{N}\left(0, \omega_b\right)$, with high probability we have that $|b_{i,0}| \leq C_4 \omega_b$ for some constant $C_4$. To bound $|b_{i,t} - b_{i,0}|$, we use the argument that the weight perturbation is small when the over-parameterization is

---

[1]In the proof of Proposition 1 in Ye et al. (2020), on page 16, the two equations after Equation (16) use an incorrect unrolling of the iterative upper bound.

large, as follows. We first need to upper-bound the gradient. To start, the gradient with respect to $\mathbf{b}_i$ can be written as:

$$\frac{\partial \mathcal{L}\left[f\left(\cdot, \boldsymbol{\Theta}_t\right)\right]}{\partial b_i} = \sum_{j=1}^{m} \left(f\left(\mathbf{x}_j, \boldsymbol{\Theta}_t\right) - y_j\right) \sigma_+ \left(\mathbf{a}_{i,t}^\top \mathbf{x}_j\right).$$

By the choice of our activation function, we have $\left|\sigma_+\left(\mathbf{a}_{i,t}^\top \mathbf{x}_j\right)\right| \leq 1$, and further:

$$\left|\frac{\partial \mathcal{L}\left[f\left(\cdot, \boldsymbol{\Theta}_t\right)\right]}{\partial b_i}\right| \leq \sum_{j=1}^{m} |f\left(\mathbf{x}_j, \boldsymbol{\Theta}_t\right) - y_j| \leq \sqrt{m}\, \|\hat{\mathbf{y}}_t - \mathbf{y}\|_2 = \sqrt{m}\mathcal{L}\left[f\left(\cdot, \boldsymbol{\Theta}_t\right)\right]^{\frac{1}{2}}.$$

This implies that

$$|b_{i,t} - b_{i,0}| = \sum_{\tau=0}^{t-1} |b_{i,\tau+1} - b_{i,\tau}| \leq \eta \sum_{\tau=0}^{t-1} \left|\frac{\partial \mathcal{L}\left[f\left(\cdot, \boldsymbol{\Theta}_t\right)\right]}{\partial b_i}\right| \leq \eta \sqrt{m} \sum_{\tau=0}^{t-1} \mathcal{L}\left[f\left(\cdot, \boldsymbol{\Theta}_\tau\right)\right]^{\frac{1}{2}}.$$

By Theorem 7, we have that:

$$\sum_{\tau=0}^{t-1} \mathcal{L}\left[f\left(\cdot, \boldsymbol{\Theta}_\tau\right)\right]^{\frac{1}{2}} \leq \mathcal{L}\left[f\left(\cdot, \boldsymbol{\Theta}_0\right)\right]^{\frac{1}{2}} \sum_{t=0}^{\tau-1} \left(1 - C_1 \eta N \lambda_{\min}^2\right)^{\frac{\tau}{2}}$$

$$\leq \mathcal{L}\left[f\left(\cdot, \boldsymbol{\Theta}_0\right)\right]^{\frac{1}{2}} \sum_{t=0}^{\tau-1} \left(1 - \frac{1}{2} C_1 \eta N \lambda_{\min}^2\right)^{\tau}$$

$$\leq \frac{2}{C_1 \eta N \lambda_{\min}^2} \mathcal{L}\left[f\left(\cdot, \boldsymbol{\Theta}_0\right)\right]^{\frac{1}{2}}$$

Thus we have:

$$|b_{i,t} - b_{i,0}| \leq \eta \sqrt{m} \cdot \frac{2}{C_1 \eta N \lambda_{\min}^2} \mathcal{L}\left[f\left(\cdot, \boldsymbol{\Theta}_0\right)\right]^{\frac{1}{2}} = \frac{2\sqrt{m}}{C_1 N \lambda_{\min}^2} \mathcal{L}\left[f\left(\cdot, \boldsymbol{\Theta}_0\right)\right]^{\frac{1}{2}}$$

When $N = \Omega\left(\frac{m^4}{\lambda_{\min}^8} \mathcal{L}\left[f\left(\cdot, \boldsymbol{\Theta}_0\right)\right]^2\right)$, we have that $\frac{2\sqrt{m}}{C_1 N \lambda_{\min}^2} \mathcal{L}\left[f\left(\cdot, \boldsymbol{\Theta}_0\right)\right]^{\frac{1}{2}} \leq C_2 m^{-\frac{1}{2}} N^{-\frac{3}{4}}$ for some constant $C_2 > 0$. Therefore, with high probability, $|b_{i,t}|$ can be bounded as:

$$|b_{i,t}| \leq |b_{i,0}| + |b_{i,t} - b_{i,0}| \leq C_4 \omega_b + C_2 m^{-\frac{1}{2}} N^{-\frac{3}{4}} \leq \left(C_2 + C_4\right) m^{-\frac{1}{2}} N^{-\frac{3}{4}} \leq C_3 m^{-\frac{1}{2}} N^{-\frac{3}{4}},$$

by letting $C_3 = C_2 + C_4$. Plugging the bound of $|b_{i,t}|$ into (17), we have:

$$\|\boldsymbol{\Phi}_i\|_2 \leq \sqrt{m} N \cdot C_3 m^{-\frac{1}{2}} N^{-\frac{3}{4}} = C_3 N^{\frac{1}{4}}.$$

$\square$

Since each $b_{i,0}$ are initialized with zero-mean, we must have that $\mathbb{E}_{\boldsymbol{\Theta}_0}\left[f\left(\cdot, \boldsymbol{\Theta}_0\right)\right] = 0$. Therefore, the expected loss at initialization can be computed as $\mathbb{E}_{\boldsymbol{\Theta}_0}\left[\mathcal{L}\left[f\left(\cdot, \boldsymbol{\Theta}_0\right)\right]\right] = \mathbb{E}_{\boldsymbol{\Theta}_0}\left[\|f\left(\mathbf{X}, \boldsymbol{\Theta}_0\right)\|_2^2\right] + \|\mathbf{y}\|_2^2$. Using the independence between $b_{i,0}$'s and the boundedness of the activation, we can compute that $\mathbb{E}_{\boldsymbol{\Theta}_0}\left[\|f\left(\mathbf{X}, \boldsymbol{\Theta}_0\right)\|_2^2\right] \leq \frac{1}{2}\sum_{j=1}^m \sum_{i=1}^n \mathbb{E}\left[b_{i,0}^2\right] \leq O(1)$ when $\omega_b \leq O\left(m^{-\frac{1}{2}} N^{-\frac{3}{4}}\right)$ as assumed in Lemma 3. This implies that with a high probability $\mathcal{L}\left[f\left(\cdot, \boldsymbol{\Theta}_0\right)\right] = O(1)$ and the overparameterization requirement reduces to $N = \Omega\left(\frac{m^4}{\lambda_{\min}^8}\right)$. With an established bound of $\|\boldsymbol{\Phi}_i\|_2$, we can provide a bound for both $\mathcal{D}_{\mathcal{M}_N}$ and $\mathcal{L}\left[f_{\mathcal{S}_1}\left(\cdot, \boldsymbol{\Theta}_t\right)\right]$.

**Theorem 1.** *Let Assumption 1 and Assumption 2 hold. Let $\mathcal{M}_N$ be the polytope formed by neuron outputs from the pre-trained network with gradient descent. If $N = \Omega\left(\frac{m^4}{\lambda_{\min}^8} \mathcal{L}\left[f\left(\cdot, \boldsymbol{\Theta}_0\right)\right]^2\right)$, then with high probability, we have that:*

$$\mathcal{D}_{\mathcal{M}_N} = O\left(N^{\frac{1}{4}}\right); \quad \mathcal{L}\left[f_{\mathcal{S}_1}\left(\cdot, \boldsymbol{\Theta}_t\right)\right] \leq O\left(\sqrt{N}\right).$$

*Proof.* To show the upper bound on $\mathcal{D}_{\mathcal{M}_N}$, we can directly apply Lemma 3 and Lemma 2 to get that:

$$\mathcal{D}_{\mathcal{M}_N} \leq \mathcal{D}' \leq 2 \max_{i \in [N]} \|\boldsymbol{\Phi}_i\|_2 \leq 2C_3 N^{\frac{1}{4}} = O\left(N^{\frac{1}{4}}\right).$$

To show the upper bound on $\mathcal{L}[f_{\mathcal{S}_1}(\cdot, \boldsymbol{\Theta}_t)]$, we let $i^* \in \mathcal{S}_1$ be given. Then it holds that:

$$\mathcal{L}[f_{\mathcal{S}_1}(\cdot, \boldsymbol{\Theta}_t)] = \frac{1}{2}\|\boldsymbol{\Phi}_{i^*} - \mathbf{y}\|_2^2 \leq \frac{1}{2}(\|\boldsymbol{\Phi}_{i^*}\|_2 + \|\mathbf{y}\|_2)_2^2 \leq \|\boldsymbol{\Phi}_{i^*}\|_2^2 + \|\mathbf{y}\|_2^2.$$

By Assumption 2, we have that $\|\mathbf{y}\|_2 = 1$. Moreover, by Lemma 3 we have that $\|\boldsymbol{\Phi}_{i^*}\|_2 \leq C_3 N^{\frac{1}{4}}$. Therefore, we have:

$$\mathcal{L}[f_{\mathcal{S}_1}(\cdot, \boldsymbol{\Theta}_t)] \leq C_3^2\sqrt{N} + 1 \leq (C_3^2 + 1)\sqrt{N} = O\left(\sqrt{N}\right).$$

$\square$

Under the setting of Theorem 1, the training loss convergence during the pruning phase in Lemma 1 becomes:

$$
\begin{aligned}
\mathcal{L}[f_{\mathcal{S}_k}(\cdot, \boldsymbol{\Theta})] &\leq \frac{1}{k}\mathcal{L}[f_{\mathcal{S}_1}(\cdot, \boldsymbol{\Theta})] + \frac{1 + \log k}{2k}\mathcal{D}_{\mathcal{M}_N}^2 + \frac{k-1}{k}\mathcal{L}[f(\cdot, \boldsymbol{\Theta})] \\
&\leq \frac{1}{k} \cdot O\left(\sqrt{N}\right) + \frac{1 + \log k}{2k} \cdot O\left(N^{\frac{1}{4}}\right)^2 + \frac{k-1}{k}\mathcal{L}[f(\cdot, \boldsymbol{\Theta})] \\
&= O\left(\frac{\log k}{k} \cdot \sqrt{N}\right) + \frac{k-1}{k}\mathcal{L}[f(\cdot, \boldsymbol{\Theta})]
\end{aligned}
$$

In this scenario, Lemma 1 shows a $O\left(\frac{\log k}{k}\right)$ decay of the loss on the pruned neural network, up to the loss achieved by the dense neural network. Additionally, because Algorithm 1 selects a single neuron during each iteration, we must have $|\mathcal{S}_k| \leq k$. In order to guarantee that $\mathcal{L}[f_{\mathcal{S}_k}(\cdot, \boldsymbol{\Theta})] \leq \epsilon + \mathcal{L}[f(\cdot, \boldsymbol{\Theta})]$ for some $\epsilon > 0$, we enforce $\frac{\log k}{k} \cdot \sqrt{N} \leq \epsilon$ to obtain that $k = O\left(\frac{\sqrt{N}}{\epsilon}\left|W_0\left(-\frac{\epsilon}{\sqrt{N}}\right)\right|\right)$ where $W_0(\cdot)$ is the Lamber $W$ function. Further applying that $W_0(x) \leq \log x$ gives that the resulting subnetwork satisfies the sparsity constraint $|\mathcal{S}_k| = \mathcal{O}\left(\frac{\sqrt{N}}{\epsilon}\log\frac{\sqrt{N}}{\epsilon}\right)$. This shows that greedy forward selection can obtain an error close to $\mathcal{L}[f(\cdot, \boldsymbol{\Theta})]$ in a with a small sparsity. In the next section, we proceed to investigate how $\mathcal{L}[f(\cdot, \boldsymbol{\Theta})]$ evolves during the pre-training phase.

## 4.2 Connecting with Pre-training Loss Convergence

By leveraging the recently developed result of neural network training convergence (Liu et al., 2021), we can extend Lemma 1 with the following upper bound on the dense neural network loss after pre-training. To state the result, we first define the network-related quantity $\lambda_{\min}$ and $\lambda_{\max}$ as follows

**Definition 1.** *Consider the first-layer output matrix at initialization $\boldsymbol{\Psi} \in \mathbb{R}^{N \times m}$ given by $\boldsymbol{\Psi}_{ij} = \sigma_+\left(\mathbf{a}_{i,0}^\top\mathbf{x}_j\right)$. We define $\lambda_{\min} = \frac{1}{\sqrt{N}}\sigma_{\min}(\boldsymbol{\Psi})$ and $\lambda_{\max} = \frac{1}{\sqrt{N}}\sigma_{\max}(\boldsymbol{\Psi})$.*

It is hard to provide a precise lower bound on $\lambda_{\min}$, but we can estimate its scale using the following reasoning. Since $\mathbf{a}_{i,0} \sim \mathcal{N}\left(0, \omega_a^2\mathbf{I}_d\right)$ are Gaussian random vectors, the pre-activation values $\mathbf{a}_{i,0}^\top\mathbf{x}_j$ follows a Gaussian distribution. Moreover, the activation function $\sigma_+(\cdot)$ further "squeezes" the value into the interval $[-1, 1]$. Therefore, when $\omega_a$ is large enough, $\sigma_+\left(\mathbf{a}_{i,0}^\top\mathbf{x}_j\right)$ behaves similar to a Gaussian distribution with constant variance. Since $\boldsymbol{\Psi} \in \mathbb{R}^{N \times m}$, standard random matrix theory shows that $\sigma_{\min}(\boldsymbol{\Psi}) = \Theta\left(\sqrt{N}\right)$ with high probability when $m$ is fixed. Therefore, we can estimate that $\lambda_{\min} = \Theta(1)$. In Appendix C we provide experimental results to verify that $\sigma_{\min}(\boldsymbol{\Psi}) = \Theta\left(\sqrt{N}\right)$.

**Theorem 2.** *Let Assumption 1 and Assumption 2 hold with $\omega_a\omega_b \leq O\left(\frac{1}{\sqrt{dN}}\right)$ where $d$ is the inpu dimension, and that a two-layer network of width $N = \Omega\left(\frac{m^4}{\lambda_{\min}^8}\mathcal{L}[f(\cdot, \boldsymbol{\Theta}_0)]^2\right)$ was pre-trained for $t$ iterations with gradient*

*descent in (5) over a dataset $D$ of size $m$.[2] Let $\mathbf{\Theta}_0$ denote the weights at initialization. If we choose the step size $\eta = O\left(\frac{1}{\sqrt{m}\left(1+\mathcal{L}[f(\cdot,\mathbf{\Theta}_0)]^{\frac{1}{2}}\right)+N\lambda_{\max}^2}\right)$, then the sequence $\{\mathcal{S}_k\}_{k=1}^{\infty}$ generated by applying Algorithm 1 to the pre-trained network satisfies:*

$$\mathcal{L}\left[f_{\mathcal{S}_k}(\cdot, \mathbf{\Theta}_t)\right] \leq O\left(\frac{\log k}{k}\cdot\sqrt{N}\right) + \left(1 - C_1\eta N\lambda_{\min}^2\right)^t \mathcal{L}\left[f(\cdot,\mathbf{\Theta}_0)\right],$$

*for some constant $C_1 > 0$.*

Since the proof only involves applying the existing characterization of the training convergence in Song et al. (2021) to the bound in Lemma 1, we defer the proof to Appendix B. Notice that the loss in Theorem 2 only decreases during successive pruning iterations if the rightmost term does not dominate the expression, i.e., all other terms decay as $O\left(\frac{\log k}{k}\right)$. If this term does not decay with increasing $k$, the upper bound on subnetwork training loss deteriorates, thus eliminating any guarantees on subnetwork performance. Interestingly, we discover that the tightness of the upper bound in Theorem 2 depends on the number of dense networks pre-training iterations.

**Theorem 3.** *Adopting identical assumptions and notation as Theorem 2, assume the dense network is pruned via greedy forward selection for $k$ iterations. The resulting subnetwork is guaranteed to achieve a training loss $O\left(\frac{\log k}{k}\cdot\sqrt{N}\right)$ if $t$—the number of gradient descent pre-training iterations on the dense network—satisfies the following condition:*

$$t \gtrapprox O\left(\frac{-\log k}{\log\left(1 - C_1\eta N\lambda_{\min}^2\right)}\right), \quad \text{where } C_1 \text{ is a positive constant.} \tag{18}$$

*Otherwise, the loss of the pruned network is not guaranteed to improve over successive iterations of greedy forward selection.*

*Proof.* From Theorem 3, one can observe that the term $O\left(\frac{\log k}{k}\cdot\sqrt{N}\right)$ dominates only when:

$$\frac{\log k}{k}\cdot\sqrt{N} \geq \left(1 - C_1\eta N\lambda_{\min}^2\right)^t \mathcal{L}\left[f(\cdot,\mathbf{\Theta}_0)\right].$$

Therefore, to guarantee that $O\left(\frac{\log k}{k}\cdot\sqrt{N}\right)$ dominates, $t$ must satisfy:

$$t \geq \frac{\log\left(\frac{\sqrt{N}\log k}{k\mathcal{L}[f(\cdot,\mathbf{\Theta}_0)]}\right)}{\log\left(1 - C_1\eta N\lambda_{\min}^2\right)} \geq \frac{-\log k - \log\mathcal{L}\left[f(\cdot,\mathbf{\Theta}_0)\right]}{\log\left(1 - C_1\eta N\lambda_{\min}^2\right)} = O\left(\frac{-\log k}{\log\left(1 - C_1\eta N\lambda_{\min}^2\right)}\right),$$

since $\mathcal{L}\left[f(\cdot,\mathbf{\Theta}_0)\right]$ is independent of $k$. $\qquad\square$

Theorem 3 states that *a threshold exists in the number of GD pre-training iterations of a dense network, beyond which subnetworks derived with greedy selection are guaranteed to achieve good training loss.* Denoting this threshold as $t^{\star}$. Consider the choice of $\eta$ in Theorem 2. When $N \gg m$, the step size becomes $\eta = O\left(\frac{1}{N\lambda_{\max}^2}\right)$ and in this case $t^{*} = O\left(\frac{-\log k}{\log(1-C_1\cdot\lambda_{\min}^2/\lambda_{\max}^2)}\right)$. This implies that the threshold of the pre-training steps depends on $\frac{\lambda_{\max}}{\lambda_{\min}}$, the condition number of $\mathbf{\Psi}$. Since $\mathbf{\Psi}$ is close to a random Gaussian matrix, $\frac{\lambda_{\max}}{\lambda_{\min}}$ increases as the size of the dataset $m$ increases. This implies that larger datasets require more pre-training for discovering high-performing subnetworks. Our finding provides theoretical insight regarding $i$) how much pre-training is sufficient to discover a well-performing subnetwork and $ii$) the difficulty of discovering high-performing subnetworks in large-scale experiments (i.e., large datasets require more pre-training).

**Remark.** The combination of the initialization scale requirement in Lemma 3 and Theorem 2 puts our training scheme in the so-called NTK regime (Song et al., 2021), where the first-layer weights change only

---

[2]This overparameterization assumption is mild in comparison to previous work (Allen-Zhu et al., 2018; Du et al., 2019).

slightly, resulting in a training dynamic similar to the random feature model. While the training in this regime prevents the model from learning useful features and thus results in poor generalization, our initialization is crucial to control the convex hull diameter $\mathcal{D}_{\mathcal{M}_N}$'s in a reasonable magnitude. However, it should be expected that this requirement can be relaxed if one assumes additional structure on the training data $\mathbf{X}$: a sufficiently trained neural network should map the input into one or multiple clusters with small diameters. Future works can consider performing a more fine-grained analysis of the relationship between the amount of pre-training, the structure of the data, and the diameter of the hidden output of the pre-trained network.

## 5 Related work

Many variants have been proposed for both structured (Han et al., 2016; Li et al., 2016; Liu et al., 2017; Ye et al., 2020) and unstructured (Evci et al., 2019; 2020; Frankle & Carbin, 2018; Han et al., 2015) pruning. Generally, structured pruning, which prunes entire channels or neurons of a network instead of individual weights, is considered more practical, as it can achieve speedups without leveraging specialized libraries for sparse computation. Existing pruning criterion include the norm of weights (Li et al., 2016; Liu et al., 2017), feature reconstruction error (He et al., 2017; Luo et al., 2017; Ye et al., 2020; Yu et al., 2017), or even gradient-based sensitivity measures (Baykal et al., 2019; Wang et al., 2020; Zhuang et al., 2018). While most pruning methodologies perform backward elimination of neurons within the network (Frankle & Carbin, 2018; Frankle et al., 2019; Liu et al., 2017; 2018; Yu et al., 2017), some recent research has focused on forward selection structured pruning strategies (Ye et al., 2020; Ye et al., 2020; Zhuang et al., 2018). *We adopt greedy forward selection within this work, as it has been previously shown to yield superior performance in comparison to greedy backward elimination, and it is a simple algorithm to apply.*

Empirical analysis of pruning techniques has inspired associated theoretical developments. Several works have derived bounds for the performance and size of subnetworks discovered in randomly-initialized networks (Malach et al., 2020; Orseau et al., 2020; Pensia et al., 2020). Other theoretical works analyze pruning via greedy forward selection (Ye et al., 2020; Ye et al., 2020). In addition to enabling analysis concerning subnetwork size, pruning via greedy forward selection was shown to work well in practice for large-scale architectures and datasets. Some findings from these works apply to randomly-initialized networks given proper assumptions (Ye et al., 2020; Malach et al., 2020; Orseau et al., 2020; Pensia et al., 2020), *but, to the best of our knowledge, no work yet analyzes how different levels of pre-training impact the performance of pruned networks from a theoretical perspective.*

*Our analysis resembles that of the Frank-Wolfe algorithm (Frank et al., 1956; Jaggi, 2013), a widely-used and simple technique for constrained, convex optimization.* Recent work has shown that training deep networks with Frank-Wolfe can be feasible in certain cases despite the non-convex nature of neural network training (Bach, 2014; Pokutta et al., 2020). Instead of training networks from scratch with Frank-Wolfe, we use a Frank-Wolfe-style approach to greedily select neurons from a pre-trained model. Such a formulation casts structured pruning as convex optimization over a marginal polytope, which can be analyzed similarly to Frank-Wolfe (Ye et al., 2020; Ye et al., 2020) and loosely approximates networks trained with standard, gradient-based techniques (Ye et al., 2020). Several distributed variants of the Frank-Wolfe algorithm have been analyzed theoretically (Wai et al., 2017; Xian et al., 2021; Hou et al., 2022), though our analysis most closely resembles that of (Bellet et al., 2015). Alternative analysis methods for greedy selection algorithms could also be constructed using sub-modular optimization techniques (Nemhauser et al., 1978).

Much work has been done to analyze the convergence properties of neural networks trained with gradient-based techniques (Chang et al., 2020; Hanin & Nica, 2019; Jacot et al., 2018; Zhang et al., 2019). Such convergence rates were initially explored for wide, two-layer neural networks using mean-field analysis techniques (Mei et al., 2019; 2018). Similar techniques later extended such analysis to deeper models (Lu et al., 2020; Xiong et al., 2020). Generally, recent work on neural network training analysis has led to novel analysis techniques (Hanin & Nica, 2019; Jacot et al., 2018), extensions to alternate optimization methodologies (Jagatap & Hegde, 2018; Oymak & Soltanolkotabi, 2019), and even generalizations to different architectural components (Goel et al., 2018; Li & Yuan, 2017; Zhang et al., 2019). By adopting such an analysis, we aim to bridge the gap between the theoretical understanding of neural network training and LTH.

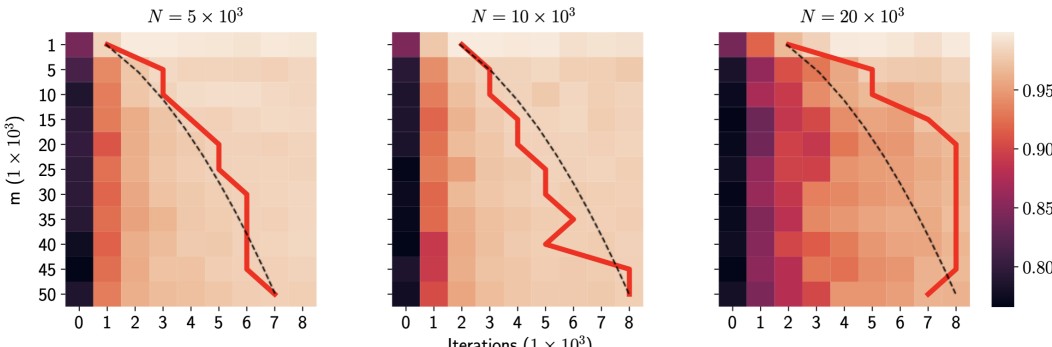

Figure 1: Pruned, two-layer models on MNIST. Sub-plots depict dense network sizes, while the $x$ and $y$ axes depict the number of pre-training iterations and the sub-dataset size. Color represents training accuracy, and the red line depicts the point at which subnetworks surpass the performance of the best-pruned model on the full dataset for different sub-dataset sizes. Dashed lines are the theoretical predicted result computed based on the condition number of the random Gaussian matrices.

## 6 Experimental Results

### 6.1 Empirical Validation on Two-Layer Networks

In this section, we empirically validate our theoretical results. Pruning via greedy forward selection has already been empirically analyzed in previous work. Therefore, *we focus on an in-depth analysis of the scaling properties of greedy forward selection with respect to the size and complexity of the underlying dataset.* This experimental setup will better support our theoretical result in Section 4.2, which predicts that larger datasets require more pre-training for subnetworks obtained via greedy forward selection to perform well. Experiments are run on an internal cluster with two Nvidia RTX 3090 GPUs using the public implementation of greedy forward selection (Ye, 2021).

We perform structured pruning experiments with two-layer networks on MNIST (Deng, 2012) by pruning hidden neurons via greedy forward selection. To match the single output neuron setup described in Section 1, we binarize MNIST labels by considering all labels less than five as zero and vice versa. Our model architecture matches the description in Section 1 with a few minor differences. Namely, we adopt a ReLU hidden activation and apply a sigmoid output transformation to enable training with binary cross-entropy loss. Experiments are conducted with several different hidden dimensions (i.e., $N \in \{5K, 10K, 20K\}$).

To study how dataset size affects subnetwork performance, we construct sub-datasets of sizes 1K to 50K (i.e., in increments of 5K) from the original MNIST dataset by uniformly sampling examples from the ten original classes. The two-layer network is pre-trained for 8K iterations in total and pruned every 1K iterations to a size of 200 hidden nodes. After pruning, the accuracy of the pruned model over the entire training dataset is recorded (i.e., no fine-tuning is performed), allowing the impact of dataset size and pre-training length on subnetwork performance to be observed. Figure 1 shows the results averaged over three trials.

**Discussion.** The performance of pruned subnetworks in Figure 1 matches the theoretical analysis provided in Section 4 for all different sizes of two-layer networks. Namely, *as the dataset size increases, so does the amount of pre-training required to produce a high-performing subnetwork.* To see this, one can track the trajectory of the red line, which traces the point at which the accuracy of the best-performing subnetwork for the entire dataset is surpassed at each sub-dataset size. This trajectory clearly illustrates that pre-training requirements for high-performing subnetworks increase with the size of the dataset. Furthermore, this increase in the required pre-training is seemingly logarithmic, as the trajectory typically plateaus at larger dataset sizes.

Interestingly, despite using a small-scale dataset, high-performing subnetworks are never discovered at initialization, revealing that minimal pre-training is often required to obtain a good subnetwork via greedy

| Model | Dataset Size | Pruned Accuracy | | | | Dense Accuracy |
|-------|--------------|-----------------|--|--|--|----------------|
| | | 20K It. | 40K It. | 60K It. | 80K It. | |
| MobileNetV2 | 10K | 82.32 | **86.18** | 86.11 | 86.09 | 83.13 |
| | 30K | 80.19 | **87.79** | 88.38 | 88.67 | 87.62 |
| | 50K | 86.71 | 88.33 | **91.79** | 91.77 | 91.44 |
| ResNet34 | 10K | 75.29 | **85.47** | 85.56 | 85.01 | 85.23 |
| | 30K | 84.06 | 91.59 | **92.31** | 92.15 | 92.14 |
| | 50K | 89.79 | 91.34 | **94.28** | 94.23 | 94.18 |

Table 1: CIFAR10 test accuracy for subnetworks derived from dense networks with varying pre-training amounts (i.e., number of training iterations listed in the top row) and sub-dataset sizes. Numbers marked in bold denote the setting where the pruned network achieves comparable performance to the dense network in the smallest training iterations.

forward selection. Previous work claims that high-performing subnetworks may exist at initialization in theory. In contrast, our empirical analysis shows this is not the case even in simple experimental settings.

## 6.2 Application to Deeper Neural Architectures

We perform structured pruning experiments (i.e., channel-based pruning) using ResNet34 (He et al., 2015) and MobileNetV2 (Sandler et al., 2018) architectures on CIFAR10 and ImageNet (Krizhevsky et al., 2009; Deng et al., 2009). We adopt the same generalization of greedy forward selection to pruning deep networks as described in (Ye et al., 2020) and use $\epsilon$ to denote our stopping criterion. We follow the three-stage methodology—pre-training, pruning, and fine-tuning—and modify both the size of the underlying dataset and the amount of pre-training before pruning to examine their impact on subnetwork performance. Standard data augmentation and splits are adopted for both datasets.

**CIFAR10**. Three CIFAR10 sub-datasets of size 10K, 30K, and 50K (i.e., full dataset) are created using uniform sampling across classes. Pre-training is conducted for 80K iterations using SGD with momentum and a cosine learning rate decay schedule starting at 0.1. We use a batch size of 128 and weight decay of $5 \cdot 10^{-4}$.[3] The dense model is independently pruned every 20K iterations, and subnetworks are fine-tuned for 2500 iterations with an initial learning rate of 0.01 before being evaluated. We adopt $\epsilon = 0.02$ and $\epsilon = 0.05$ for MobileNet-V2 and ResNet34, respectively, yielding subnetworks with a 40% decrease in FLOPS and 20% decrease in model parameters in comparison to the dense model.[4]

The results of these experiments are presented in Table 1. The amount of training required to discover a high-performing subnetwork consistently increases with the size of the dataset. For example, with MobileNetV2, a winning ticket is discovered on the 10K and 30K sub-datasets in only 40K iterations. In comparison, for the 50K sub-dataset, a winning ticket is not found until 60K iterations of pre-training have been completed. Furthermore, subnetwork performance often surpasses the fully-trained dense network *without completing the entire pre-training procedure*.

**ImageNet**. We perform experiments on the ILSVRC2012, 1000-class dataset (Deng et al., 2009) to determine how pre-training requirements change for subnetworks pruned to different FLOP levels.[5] We adopt the same experimental and hyperparameter settings as Ye et al. (2020). Models are pre-trained for 150 epochs using SGD with momentum and cosine learning rate decay with an initial value of 0.1. We use a batch size of 128 and weight decay of $5 \cdot 10^{-4}$. The dense network is independently pruned every 50 epochs, and the subnetwork is fine-tuned for 80 epochs using a cosine learning rates schedule with an initial value of 0.01 before being evaluated. We first prune models with $\epsilon = 0.02$ and $\epsilon = 0.05$ for MobileNetV2 and ResNet34,

---

[3]Our pre-training settings are adopted from a popular repository for the CIFAR10 dataset (Liu, 2017).

[4]These settings are derived using a grid search over values of $\epsilon$ and the learning rate with performance measured over a hold-out validation set; see Appendix D.

[5]We do not experiment with different sub-dataset sizes on ImageNet due to limited computational resources.

| Model | FLOP (Param) Ratio | Pruned Accuracy | | | Dense Accuracy |
|---|---|---|---|---|---|
| | | 50 Epoch | 100 Epoch | 150 Epoch | |
| MobileNetV2 | 60% (80%) | 70.05 | 71.14 | 71.53 | 71.70 |
| | 40% (65%) | 69.23 | 70.36 | 71.10 | |
| ResNet34 | 60% (80%) | 71.68 | 72.56 | 72.65 | 73.20 |
| | 40% (65%) | 69.87 | 71.44 | 71.33 | |

Table 2: Test accuracy on ImageNet of subnetworks with different FLOP levels derived from dense models with varying amounts of pre-training (i.e., training epochs listed in the top row). We report the FLOP/parameter ratio after pruning with respect to the FLOPS/parameters of the dense model.

respectively, yielding subnetworks with a 40% reduction in FLOPS and 20% reduction in parameters in comparison to the dense model. Pruning is also performed with a larger $\epsilon$ value (i.e., $\epsilon = 0.05$ and $\epsilon = 0.08$ for MobileNetV2 and ResNet34, respectively) to yield subnetworks with a 60% reduction in FLOPS and 35% reduction in model parameters in comparison to the dense model.

The results are reported in Table 2. Although the dense network is pre-trained for 150 epochs, subnetwork test accuracy reaches a plateau after only 100 epochs of pre-training in all cases. Furthermore, subnetworks with only 50 epochs of pre-training still perform well in many cases. For example, the 60% FLOPS ResNet34 subnetwork with 50 epochs of pre-training achieves a testing accuracy within 1% of the pruned model derived from the fully pre-trained network. Thus, *high-performing subnetworks can be discovered with minimal pre-training even on large-scale datasets like ImageNet.*

**Discussion.** These results demonstrate that the number of dense network pre-training iterations needed to reach a plateau in subnetwork performance $i$) consistently increases with the size of the dataset, and $ii$) is consistent across different architectures given the same dataset. Discovering a high-performing subnetwork on the ImageNet dataset takes roughly 500K pre-training iterations (i.e., 100 epochs). In comparison, discovering a subnetwork that performs well on the MNIST and CIFAR10 datasets takes roughly 8K and 60K iterations, respectively. Thus, the amount of required pre-training iterations increases based on the size of dataset *even across significantly different scales and domains*. This indicates that the dependence of pre-training requirements on dataset size may be an underlying property of discovering high-performing subnetworks no matter the experimental setting.

Interestingly, we observe that dense network size does impact subnetwork performance. In Figure 1, subnetwork performance varies based on dense network width, and subnetworks derived from narrower dense networks seem to achieve better performance. Similarly, in Tables 1 and 2, subnetworks derived from MobileNetV2 tend to achieve higher relative performance than the dense model. Thus, subnetworks derived from smaller dense networks seem to achieve better *relative* performance in comparison to those derived from larger dense networks, suggesting that pruning via greedy forward selection may demonstrate different qualities in contrast to more traditional approaches (e.g., iterative magnitude-based pruning (Liu et al., 2018)). Despite this observation, however, the amount of pre-training epochs required for the emergence of the best-performing subnetwork is still consistent across architectures and dependent on dataset size.

**Connection to Theoretical Results.** The experimental result on deep neural networks is consistent with the intuition that our theoretical results may also generalize to deeper architectures. In particular, notice that our theory consists of two components: $i$). an upper bound on the pruning error, and $ii$).a convergence guarantee that depends on the minimum eigenvalue of the NTK matrix. For $i$), we should note that, since the algorithm for deep neural networks involves a layer-wise pruningYe et al. (2020), our analysis has the potential to extend to deeper architecture as long as one controls the error in the output of each intermediate layer from pruning weights in the previous layers and treat this output as the input to the following layers. For $ii$), the emerging analysis on the convergence property of training deep neural networks Liu et al. (2023; 2021); Nguyen (2021) shed light on some theoretical evidence. In particular, it is shown in Nguyen (2021);

Nguyen et al. (2022) that the condition number of the NTK of the deep ReLU network also depends on the number of training samples, which may explain why our theory could also generalize to deeper architectures.

## 7 Conclusion

In this work, we theoretically analyze the impact of dense network pre-training on the performance of a pruned subnetwork obtained via greedy forward selection. By expressing pruned network loss in terms of the number of gradient descent iterations performed on its associated dense network, we discover a threshold in the number of pre-training iterations beyond which a pruned subnetwork achieves good training loss. Our theoretical result implies this threshold's dependency on the dataset's size, which offers intuition into the early-bird ticket phenomenon and the difficulty of replicating pruning experiments at scale. We also empirically verify our theoretical findings over several datasets and network architectures, showing that the amount of pre-training required to discover a winning ticket is consistently dependent on the size of the underlying dataset. Beyond the materials in the main text, we also included in Appendix E a distributed version of the greedy forward selection algorithm and its empirical performance evaluation.

Several problems remain, such as extending our analysis beyond two-layer networks, deriving generalization bounds for subnetworks pruned with greedy forward selection, or even using our theoretical results to discover new heuristic methods for identifying early-bird tickets in practice. Moreover, future works can consider extending our theory to deeper architectures. In particular, a generalization of Algorithm 1 is given in (Ye et al., 2020). To apply our analysis to a deeper architecture, future work should consider performing a layer-wise analysis on the convergence of the pruning algorithm and upper-bound the propagation of the errors through layers. The pruning algorithm we consider also has the potential to extend to other architectures. For example, we provided Algorithm 2 as a generalization of Algorithm 1 to CNN.

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

## A  Proof of Lemma 2

*Proof.* We will proceed with the proof by induction on the scope of $\boldsymbol{\Phi}_i$'s that $\mathbf{u}$ and $\mathbf{v}$ are composed of. To be more specific, we will show that $\|\mathbf{u} - \mathbf{v}\|_2 \leq \mathcal{D}'$ for all $\mathbf{u}, \mathbf{v} \in \mathtt{Conv}\{\boldsymbol{\Phi}_i; i \in [n]\}$ for $n = 1, \dots, N$. For the base case, we consider $n = 1$. We must have that $\mathbf{u} = \boldsymbol{\Phi}_1 = \mathbf{v}$. Therefore, $\|\mathbf{u} - \mathbf{v}\|_2 = 0 \leq \mathcal{D}'$. For the inductive case, suppose that our claim is true for $n = 1, \dots, n'$. We shall prove that it is true for $n = n' + 1$. In this case,

$$\mathbf{u} = \sum_{i=1}^{n'+1} \gamma_i \boldsymbol{\Phi}_i; \quad \mathbf{v} = \sum_{i=1}^{n'+1} \gamma_i' \boldsymbol{\Phi}_i$$

for some $\{\gamma_i\}_{i=1}^{n'+1}$ and $\{\gamma_i'\}_{i=1}^{n'+1}$ satisfying $\sum_{i=1}^{n'+1} \gamma_i = \sum_{i=1}^{n'+1} \gamma_i' = 1$ and $\gamma_i \geq 0, \gamma_i' \geq 0$ for all $i \in [n'+1]$. Without loss of generality, let $\gamma_{n'+1} \geq \gamma_{n'+1}'$. If $\gamma_{n'+1} = 1$, then we have

$$
\begin{aligned}
\|\mathbf{u} - \mathbf{v}\|_2 &= \left\| \boldsymbol{\Phi}_{n'+1} - \sum_{i=1}^{n'+1} \gamma_i' \boldsymbol{\Phi}_i \right\|_2 \\
&= \left\| \sum_{i=1}^{n'} \gamma_i' \left( \boldsymbol{\Phi}_{n'+1} - \boldsymbol{\Phi}_i \right) \right\|_2 \\
&\leq \sum_{i=1}^{n'} \gamma_i' \left\| \boldsymbol{\Phi}_{n'+1} - \boldsymbol{\Phi}_i \right\|_2 \\
&\leq \mathcal{D}' \sum_{i=1}^{n'} \gamma_i' \\
&\leq \mathcal{D}'
\end{aligned}
$$

Otherwise, we can suppose $\gamma_{n'+1}' \leq \gamma_{n'+1} < 1$. In this case, we can write $\mathbf{u}$ and $\mathbf{v}$ as

$$\mathbf{u} = \gamma_{n'+1} \boldsymbol{\Phi}_{n'+1} + (1 - \gamma_{n'+1}) \sum_{i=1}^{n'} \frac{\gamma_i}{1 - \gamma_{n'+1}} \boldsymbol{\Phi}_i = \gamma_{n'+1} \boldsymbol{\Phi}_{n'+1} + (1 - \gamma_{n'+1}) \mathbf{u}'$$

$$\mathbf{v} = \gamma_{n'+1}' \boldsymbol{\Phi}_{n'+1} + \left( 1 - \gamma_{n'+1}' \right) \sum_{i=1}^{n'} \frac{\gamma_i'}{1 - \gamma_{n'+1}'} \boldsymbol{\Phi}_i = \gamma_{n'+1}' \boldsymbol{\Phi}_{n'+1} + \left( 1 - \gamma_{n'+1}' \right) \mathbf{v}'$$

for some $\mathbf{u}', \mathbf{v}' \in \mathtt{Conv}\{\boldsymbol{\Phi}_i : i \in [n']\}$. Then by the inductive hypothesis, we have $\|\mathbf{u}' - \mathbf{v}'\|_2 \leq \mathcal{D}'$. Thus we have

$$
\begin{aligned}
\|\mathbf{u} - \mathbf{v}\|_2 &= \left\| \left( \gamma_{n'+1} - \gamma_{n'+1}' \right) \left( \boldsymbol{\Phi}_{n'+1} - \mathbf{v}' \right) + (1 - \gamma_{n'+1}) \left( \mathbf{u}' - \mathbf{v}' \right) \right\|_2 \\
&\leq \left( \gamma_{n'+1} - \gamma_{n'+1}' \right) \left\| \boldsymbol{\Phi}_{n'+1} - \mathbf{v}' \right\|_2 + (1 - \gamma_{n'+1}) \left\| \mathbf{u}' - \mathbf{v}' \right\|_2 \\
&\leq \left( \gamma_{n'+1} - \gamma_{n'+1}' \right) \left\| \sum_{i=1}^{n'} \frac{\gamma_i'}{1 - \gamma_{n'+1}'} \left( \boldsymbol{\Phi}_{n'+1} - \mathbf{v}' \right) \right\|_2 + (1 - \gamma_{n'+1}) \mathcal{D}' \\
&\leq \left( \gamma_{n'+1} - \gamma_{n'+1}' \right) \sum_{i=1}^{n'} \frac{\gamma_i'}{1 - \gamma_{n'+1}'} \left\| \boldsymbol{\Phi}_{n'+1} - \mathbf{v}' \right\|_2 + (1 - \gamma_{n'+1}) \mathcal{D}' \\
&\leq \left( \gamma_{n'+1} - \gamma_{n'+1}' \right) \mathcal{D}' \sum_{i=1}^{n'} \frac{\gamma_i'}{1 - \gamma_{n'+1}'} + (1 - \gamma_{n'+1}) \mathcal{D}' \\
&= \left( 1 - \gamma_{n'+1}' \right) \mathcal{D}' \\
&\leq \mathcal{D}'
\end{aligned}
$$

This finishes the inductive step and thus finishes the proof. $\qquad\square$

# B   Proof of Theorem 2

Our proof utilizes the result from (Song et al., 2021). We first revisit the scheme and theoretical result discussed in (Song et al., 2021). After that, we will interpret their result in the scenario we are consideration.

## B.1   Existing Result

(Song et al., 2021) considers using gradient descent to minimize the objective $h = \ell \circ \hat{f}$ where $\ell : \mathbb{R}^{\hat{d}_{\mathrm{out}}} \to \mathbb{R}$ is the loss function and $\hat{f} : \mathbb{R}^{\hat{d}_{\mathrm{in}}} \to \mathbb{R}^{\hat{d}_{\mathrm{out}}}$ is the function of the model defined over $m$ input-output pairs. In particular, (Song et al., 2021) makes the following assumption to show that gradient descent converges

**Assumption 3.** *(Gradient Descent)*

- *$\ell$ is twice differentiable, satisfies $\alpha_\ell$-PL condition, and is $\beta_\ell$-smooth.*

- *$\hat{f}$ is twice differentiable, $\beta_{\hat{f}}$-smooth.*

Building upon Assumption 3, they have the following Theorem

**Theorem 4.** *(Theorem 2 in (Song et al., 2021)). Assume that Assumption 3 holds. Let $\mathbf{w}_0 \in \mathbb{R}^{\hat{d}_{in}}$ satisfy*

$$\mu_{\hat{f}} \leq \sigma_{\min}\left(\nabla \hat{f}\left(\mathbf{w}_0\right)\right) \leq \sigma_{\max}\left(\nabla \hat{f}\left(\mathbf{w}_0\right)\right) \leq \nu_{\hat{f}}$$

*and $h(\mathbf{w}_0) = O\left(\frac{\alpha_\ell \mu_{\hat{f}}^6}{\beta_{\hat{f}}^2 \nu_{\hat{f}}^2}\right)$. Then the sequence $\{\mathbf{w}_t\}_{t=0}^\infty$ generated by*

$$\mathbf{w}_{t+1} = \mathbf{w}_t - \eta \nabla h\left(\mathbf{w}_t\right); \quad \eta = O\left(\frac{1}{\beta_{\hat{f}} \left\|\nabla \ell\left(\hat{f}\left(\mathbf{w}_0\right)\right)\right\|_2 + \beta_\ell \left(\mu_{\hat{f}}^2 + \nu_{\hat{f}}^2\right)}\right)$$

*satisfies the following convergence property*

$$h\left(\mathbf{w}_{t+1}\right) \leq \left(1 - C\eta \alpha_\ell \mu_{\hat{f}}^2\right) h\left(\mathbf{w}_t\right)$$

*for some constant $C > 0$.*

To extend this result to the training of shallow neural networks, they consider the following parameterization of the two-layer neural network and the mean-squared error loss

$$\hat{f}\left(\boldsymbol{\Theta}\right) = \mathbf{V}\sigma_+\left(\mathbf{W}\mathbf{X}\right); \quad \ell\left(\hat{\mathbf{Y}}\right) = \frac{1}{2}\left\|\hat{\mathbf{Y}} - \mathbf{Y}\right\|_F^2$$

where $\mathbf{X} \in \mathbb{R}^{d \times m}$ is the matrix consisting of the input vectors, $\mathbf{Y} \in \mathbb{R}^{d' \times m}$ is the matrix consisting of the label vectors, $\mathbf{W} \in \mathbb{R}^{N \times d}$ is the first layer weights, $\mathbf{V} \in \mathbb{R}^{d' \times N}$ is the second layer weights, $\boldsymbol{\Theta} = \{\mathbf{W}, \mathbf{V}\}$ is the collection of weights, and $\sigma_+\left(\cdot\right)$ is the entry-wise activation function. Within this setup, they make the following assumption about the neural network

**Assumption 4.** *(Neural Network)*

- *At initialization, the entries of the weight satisfies $\mathbf{W}_{ij} \sim \mathcal{N}\left(0, \omega_1^2\right)$ and $\mathbf{V} \sim \mathcal{N}\left(0, \omega_2^2\right)$ satisfying $\omega_1 \omega_2 = O\left(\frac{1}{\sqrt{dN}}\right)$.*

- *$\sigma_+\left(\cdot\right)$ is twice differentiable and satisfies $\max\left\{\left|\sigma_+'\left(\cdot\right)\right|, \left|\sigma_+''\left(\cdot\right)\right|\right\} \leq \delta$*

- *There exists $r_1, r_2$ such that $\tau^{r_1}\left|\sigma_+\left(a\right)\right| \leq \left|\sigma_+\left(\tau a\right)\right| \leq \tau^{r_2}\left|\sigma_+\left(a\right)\right|$.*

- *For all $k$, it holds that $\sigma_{\max}\left(\mathbf{V}_k\right) = O(1)$.*[6]

---

[6]This assumption can be eliminated by assuming a larger overparameterization.

With Assumption 4, they proved the following result:

**Theorem 5.** *Suppose that Assumption 4 holds. If $N = \left(m^{\frac{3}{2}}\right)$, then with high probability over the initialization, we have*

- *(Lemma 18 in (Song et al., 2021)) $\mu_{\hat{f}} := \sigma_{\min}\left(\sigma_+(\mathbf{WX})\right) \leq \sigma_{\min}\left(\nabla\hat{f}(\mathbf{\Theta}_0)\right)$.*

- *(Lemma 18 of (Song et al., 2021)) $\nu_{\hat{f}} := c_0\delta\sigma_{\max}(\mathbf{X}) + \sigma_{\max}\left(\sigma_+(\mathbf{WX})\right) \geq \sigma_{\max}\left(\nabla\hat{f}(\mathbf{\Theta}_0)\right)$ for some constant $c_0 > 0$.*

- *(Lemma 18 in (Song et al., 2021)) $\beta_{\hat{f}} = c_1\delta\sigma_{\max}(\mathbf{X})$ for some constant $c_1 > 0$.*

- *(Theorem 3 in (Song et al., 2021)) $\mathbf{\Theta}_0$ satisfies $h(\mathbf{\Theta}_0) = O\left(\frac{\alpha_\ell\mu_{\hat{f}}^6}{\beta_{\hat{f}}^2\nu_{\hat{f}}^2}\right)$ with $\mu_{\hat{f}}$ and $\nu_{\hat{f}}$ defined above.*

Combining Theorem 5 and Theorem 4, they obtain a training loss convergence for the two-layer neural network.

## B.2 Proving Theorem 2

Recall that given an input $\mathbf{x}$, our neural network is defined as

$$f(\mathbf{x}, \mathbf{\Theta}) = \frac{1}{N}\sum_{i=1}^{n}\sigma(\mathbf{x}, \boldsymbol{\theta}_i) = \sum_{i=1}^{N}b_i\sigma_+\left(\mathbf{a}_i^\top\mathbf{x}\right)$$

Indeed, generalizing to a fixed matrix of input vectors $\mathbf{X}$ and outputs $\mathbf{Y}$, our neural network can be written as

$$f(\mathbf{\Theta}) = \mathbf{b}^\top\sigma_+(\mathbf{AX}) \in \mathbb{R}^{1\times m}; \quad \mathcal{L}[f(\cdot, \mathbf{\Theta})] = \frac{1}{2}\left\|f(\mathbf{\Theta}) - \mathbf{y}^\top\right\|_F^2$$

Therefore, our neural network setup is the same as (Song et al., 2021) by letting the output dimension $d' = 1$. Moreover, by assuming our Assumption 1 and 2, Assumption 4 is satisfied with $\omega_a = m^{-\frac{1}{2}}N^{-\frac{3}{4}}$ and $\omega_b = \frac{\sqrt{m}}{\sqrt{d}}$. In this way, we have $\omega_a\omega_b = d^{-\frac{1}{2}}N^{-\frac{3}{4}} = O\left(\frac{1}{\sqrt{dN}}\right)$. Thus, interpreting Theorem 5 in our setting, we have

**Theorem 6.** *Suppose that Assumption 1 and Assumption 2 holds. If $N = \left(m^{\frac{3}{2}}\right)$, then with high probability over the initialization, we have*

- $\mu_f := \sqrt{N}\lambda_{\min} \leq \sigma_{\min}\left(\nabla\hat{f}(\mathbf{\Theta}_0)\right)$.

- $\nu_f := c_0\delta\sqrt{m} + \sqrt{N}\lambda_{\max} \geq \sigma_{\max}\left(\nabla\hat{f}(\mathbf{\Theta}_0)\right)$ *for some constant $c_0 > 0$.*

- $\beta_f := c_1\delta\sqrt{m}$ *for some constant $c_1 > 0$.*

- $\mathbf{\Theta}_0$ *satisfies $\mathcal{L}[f(\cdot, \mathbf{\Theta}_0)] = O\left(\frac{\alpha_\ell\mu_f^6}{\beta_f^2\nu_f^2}\right)$ with $\mu_f$ and $\nu_f$ defined above.*

*Proof.* To start, notice that

$$\sigma_{\max}(\mathbf{X}) \leq \|\mathbf{X}\|_F = \left(\sum_{j=1}^{m}\|\mathbf{x}_j\|_2^2\right)^{\frac{1}{2}} = \sqrt{m} \tag{19}$$

For Bullet 1, we recall that $\lambda_{\min}$ is defined as $\lambda_{\min} = \frac{1}{\sqrt{N}}\sigma_{\min}(\mathbf{A}_0\mathbf{X})$ in Definition 1. Combining with the first bullet point in Theorem 5 gives Bullet 1. For Bullet 2, we recall that $\lambda_{\max}$ is defined as $\lambda_{\max} = \frac{1}{\sqrt{N}}\sigma_{\max}(\mathbf{A}_0\mathbf{X})$ in Definition 1. Combining with the second bullet point in Theorem 5 and plugging in (19) gives the desired result. For Bullet 3, we use the third bullet in Theorem 5 and plug-in (19). Lastly, the fourth bullet directly follows from Theorem 5. $\qquad\square$

With Theorem 6, we can guarantee the convergence of training loss in our scenario.

**Theorem 7.** *Suppose that Assumption 1 and Assumption 2 holds. If $N = \left(m^{\frac{3}{2}}\right)$ and $\eta = O\left(\frac{1}{\sqrt{m}\left(1+\mathcal{L}[f(\cdot,\boldsymbol{\Theta}_0)]^{\frac{1}{2}}\right)+N\lambda_{\max}^2}\right)$, then with high probability over the initialization, we have*

$$\mathcal{L}\left[f\left(\cdot,\boldsymbol{\Theta}_{t+1}\right)\right] \leq \left(1 - C\eta N\lambda_{\min}^2\right)\mathcal{L}\left[f\left(\cdot,\boldsymbol{\Theta}_t\right)\right]$$

*Proof.* We wish to apply Theorem 4. By utilizing Theorem 6, it remains to explicitly check the requirement of the step size $\eta$ and compute the convergence rate. Notice that in Theorem 4, $\eta$ is given by

$$\eta = O\left(\frac{1}{\beta_f \left\|\nabla\ell\left(f(\mathbf{w}_0)\right)\right\|_2 + \beta_\ell\left(\mu_f^2 + \nu_f^2\right)}\right)$$

By our choice of $\ell$, we have that $\left\|\nabla\ell\left(f(\mathbf{w}_0)\right)\right\|_2 = \mathcal{L}\left[f\left(\cdot,\boldsymbol{\Theta}_0\right)\right]^{\frac{1}{2}}$ and $\beta_\ell = 1$. Moreover, using $\mu_f \leq \nu_f$, we have

$$\eta = O\left(\frac{1}{\beta_f \mathcal{L}\left[f\left(\cdot,\boldsymbol{\Theta}_0\right)\right]^{\frac{1}{2}} + 2\nu_f^2}\right) = O\left(\frac{1}{\sqrt{m}\left(1+\mathcal{L}\left[f\left(\cdot,\boldsymbol{\Theta}_0\right)\right]^{\frac{1}{2}}\right)+N\lambda_{\max}^2}\right)$$

by plugging in the value of $\beta_f$ and $\nu_f$ from Theorem 6 and omitting the constants. Moreover, for the choice of $\ell$ we have $\alpha_\ell = 1$. Then, the convergence rate was reduced to

$$C\eta\alpha_\ell\mu_f^2 = C\eta N\lambda_{\min}^2$$

$\square$

Notice that a direct consequence of Theorem 7 is that

$$\mathcal{L}\left[f\left(\cdot,\boldsymbol{\Theta}_t\right)\right] \leq \left(1 - C\eta N\lambda_{\min}^2\right)^t \mathcal{L}\left[f\left(\cdot,\boldsymbol{\Theta}_0\right)\right] \tag{20}$$

When pruning is performed after $t$ iterations of gradient descent, we can substitute $\boldsymbol{\Theta}$ with $\boldsymbol{\Theta}_t$ in Lemma 1 to get that

$$\mathcal{L}\left[f_{\mathcal{S}_k}\left(\cdot,\boldsymbol{\Theta}\right)\right] \leq \frac{1}{k}\mathcal{L}\left[f_{\mathcal{S}_1}\left(\cdot,\boldsymbol{\Theta}\right)\right] + \frac{1+\log k}{2k}\mathcal{D}_{\mathcal{M}_N}^2 + \frac{k-1}{k}\mathcal{L}\left[f\left(\cdot,\boldsymbol{\Theta}_t\right)\right]$$

Then, we use $\frac{k-1}{k} \leq 1$ and simply apply the upper bound of $\mathcal{L}\left[f\left(\cdot,\boldsymbol{\Theta}_t\right)\right]$ in (20) to get that

$$\mathcal{L}\left[f_{\mathcal{S}_k}\left(\cdot,\boldsymbol{\Theta}\right)\right] \leq \frac{1}{k}\mathcal{L}\left[f_{\mathcal{S}_1}\left(\cdot,\boldsymbol{\Theta}\right)\right] + \frac{1+\log k}{2k}\mathcal{D}_{\mathcal{M}_N}^2 + \left(1 - C\eta N\lambda_{\min}^2\right)^t \mathcal{L}\left[f\left(\cdot,\boldsymbol{\Theta}_0\right)\right]$$

## C  Numerical Experimental Verification

In Figure 2 we randomly generated $\boldsymbol{\Psi}$ matrix in the following way: we first generating $\mathbf{a}_1,\ldots\mathbf{a}_N$ with $\mathbf{a}_i \sim \mathcal{N}\left(0,\frac{1}{\sqrt{d}}\mathbf{I}_d\right)$, then, we generate $\mathbf{x}_1,\ldots,\mathbf{x}_m$ with $\mathbf{x}_j \sim \mathcal{N}\left(0,\mathbf{I}_d\right)$ and normalize each $\mathbf{x}_j$. After that, we compute $\boldsymbol{\Psi}$ by $\boldsymbol{\Psi}_{ij} = \sigma_+\left(\mathbf{a}_i^\top\mathbf{x}_j\right)$ to simulate the hidden neuron output at initialization. For each $N, d$, and $m$, we generate 10 of such $\boldsymbol{\Psi}$ matrices and record their minimum singular values' mean and standard deviation. Figure 2 plots $\sigma_{\min}\left(\boldsymbol{\Psi}\right)$ for different $N, d$, and $m$. For each $d$ and $m$, we also plotted the curve $O\left(\sqrt{N}\right)$ to compare with the curve $\sigma_{\min}\left(\boldsymbol{\Psi}\right)$. We can observe that the two curves almost overlaps, implying that $\sigma_{\min}\left(\boldsymbol{\Psi}\right)$ indeed scale with $O\left(\sqrt{N}\right)$.

In Figure 3, we conduct a simple experimental verification of the condition number of standard Gaussian random matrices of shape $N \times m$, where the condition number $\kappa$ of a matrix $\mathbf{M}$ is defined as $\kappa = \frac{\sigma_{\max}(\mathbf{M})}{\sigma_{\min}(\mathbf{M})}$. We can observe that $\kappa^{-1}$ decreases as $m$ increases.

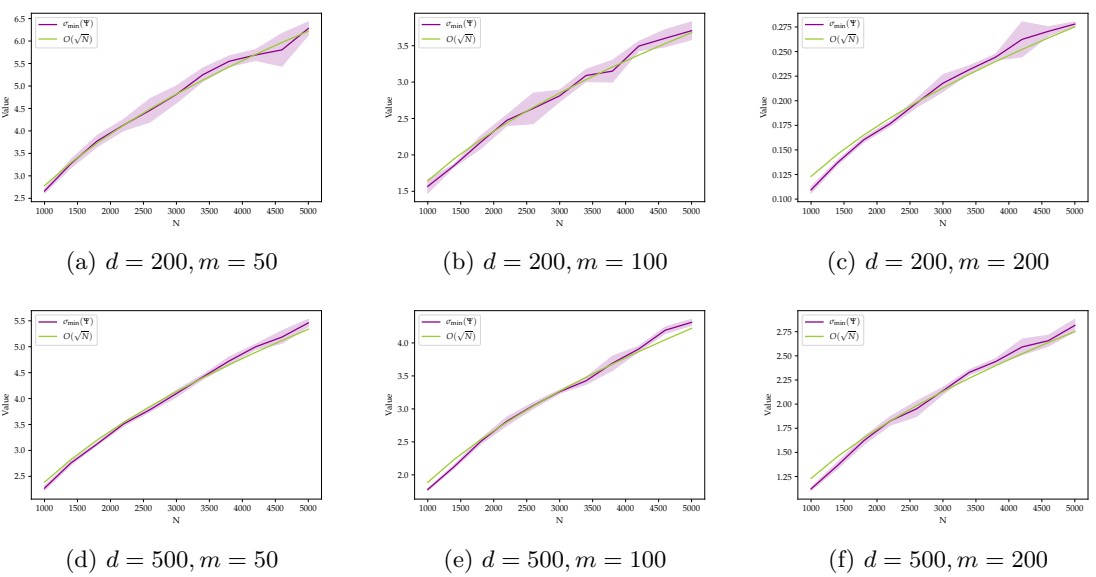

(a) $d = 200, m = 50$      (b) $d = 200, m = 100$      (c) $d = 200, m = 200$

(d) $d = 500, m = 50$      (e) $d = 500, m = 100$      (f) $d = 500, m = 200$

Figure 2: Plotting $\sigma_{\min}(\boldsymbol{\Psi})$ versus $N$ with different $d, m$. For reference, we also plotted $O(\sqrt{N})$.

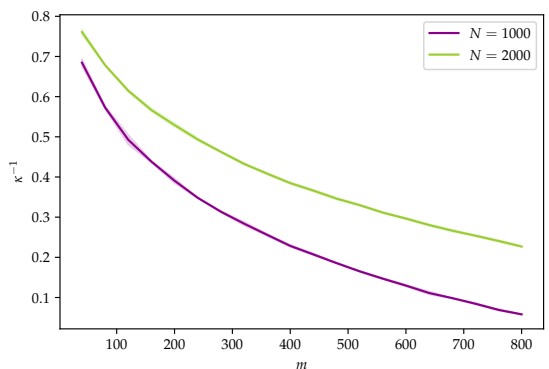

Figure 3: Inverse of the condition number of a standard Gaussian random matrix with shape $N \times m$

## D  CNN Experiments

---
**Algorithm 2** Greedy Forward Selection for Deep CNN

---
**Require:** Number of layers $L$; numbers of hidden filters $\{C_\ell\}_{\ell=1}^{L}$; training data $\mathcal{D}$; error tolerance $\epsilon$; loss function $\mathcal{L}_{\boldsymbol{\Theta}}(\mathbf{X}, \mathbf{y})$; CNN weights $\boldsymbol{\Theta}$ with $\boldsymbol{\Theta}[\ell] = \{\boldsymbol{\theta}_{\ell,j}\}_{j \in [C_\ell]}$ for $\ell \in [L]$

1: $\boldsymbol{\Theta}^{\star} := \boldsymbol{\Theta}$
2: **for** $\ell = 1, 2, \ldots, L$ **do**
3:     $\boldsymbol{\Theta}^{\star}[\ell] = \varnothing$
4:     **while** $\mathcal{L}_{\boldsymbol{\Theta}^{\star}}(\mathbf{X}, \mathbf{y}) \leq \epsilon$ **do**
5:         $\mathbf{X}, \mathbf{y} \sim \mathcal{D}$
6:         $j^* = \arg\min_{j \in [C_\ell]} \mathcal{L}_{[\boldsymbol{\Theta}^{\star}[1], \ldots, \boldsymbol{\Theta}^{\star}[\ell] \cup \{\boldsymbol{\theta}_{\ell,j}\}, \ldots, \boldsymbol{\Theta}^{\star}[C_\ell]]}(\mathbf{X}, \mathbf{y})$
7:         $\boldsymbol{\Theta}^{\star}[\ell] = \boldsymbol{\Theta}^{\star}[\ell] \cup \{j^*\}$
8:     **end while**
9: **end for**
10: **return** $\boldsymbol{\Theta}^{\star}$

---

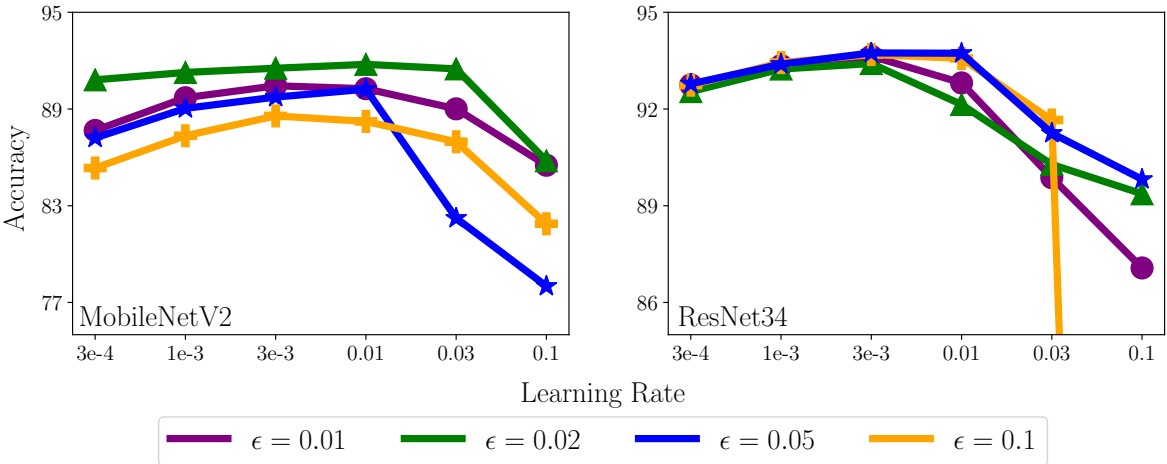

Figure 4: Subnetwork validation accuracy on the CIFAR10 dataset for different settings of $\epsilon$ and initial learning rate for fine-tuning. All models are pre-trained identically for 200 epochs. Fine-tuning is performed for 80 epochs, and we report validation accuracy for each subnetwork at the end of fine-tuning.

We begin with an in-depth algorithmic description of the greedy forward selection algorithm used for structured, channel-based pruning of multi-layer CNN architectures. This algorithm is identical to the greedy forward selection algorithm adopted in Ye et al. (2020). In this algorithm, we denote the weights of the deep network as $\boldsymbol{\Theta}$, and reference the weights within layer $\ell$ of the network as $\boldsymbol{\Theta}_\ell$. Similarly, we use $\boldsymbol{\Theta}_{\ell:}$ to denote the weights of all layers following layer $\ell$ and $\boldsymbol{\Theta}_{:\ell}$ to represent the weights of all layers up to and including layer $\ell$. $C$ denotes a list of hidden sizes within the network, where $C_\ell$ indicates the number of channels within layer $\ell$ of the CNN. Again, we use $f(\boldsymbol{\Theta}, \boldsymbol{X'})$ to denote the output of a two-layer network with parameters $\boldsymbol{\Theta}$ over the mini-batch $\boldsymbol{X'}$ and $f_\mathcal{S}(\boldsymbol{\Theta}, \boldsymbol{X'})$ to denote the same output only considering the channel indices included within the set $\mathcal{S}$.

Now, we present more details regarding the hyperparameters utilized in large-scale experiments. For ImageNet experiments, we adopt the settings of Ye et al. (2020).[7] For CIFAR10, however, we tune the setting of $\epsilon$ and the initial learning rate for fine-tuning using a grid search for both MobileNetV2 and ResNet34 architectures. This grid search is performed using a validation set on CIFAR10, constructed using a random 80-20 split on the training dataset. Optimal hyperparameters are selected based on their performance on the validation set. The results of this grid search are shown in Figure 4. As can be seen, for MobileNetV2, the best results are achieved using a setting of $\epsilon = 0.02$, which results in a subnetwork with 60% of the FLOPS of the dense model. Furthermore, an initial learning rate of 0.01 yields consistent subnetwork performance for MobileNetV2. For ResNet34, a setting of $\epsilon = 0.05$ yields the best results and a subnetwork with 60% of the FLOPS of the dense model. Again, an initial learning rate of 0.01 for fine-tuning yields the best results for ResNet34. For the rest of the hyperparameters used within CIFAR10 experiments (i.e., those used during pre-training), we adopt the settings of a widely used, open-source repository that achieves good performance on CIFAR10

## E    Distributed Greedy Forward Selection

We propose a distributed variant of greedy forward selection that can parallelize and accelerate the pruning process across multiple compute sites. Distributed greedy forward selection is shown to achieve identical theoretical guarantees compared to the centralized variant and is used to accelerate experiments with greedy forward selection within this work.

---

[7]We adopt the same experimental settings but decrease the number of fine-tuning epochs from 150 to 80 because we find that testing accuracy reaches a plateau well before 150 epochs.

For the distributed variant of greedy forward selection, we consider local compute nodes $\mathcal{V} = \{v_i\}_{i=1}^{V}$, which communicate according to an undirected connected graph $G = (\mathcal{V}, \mathcal{E})$. Here, $\mathcal{E}$ is a set of edges, where $|\mathcal{E}| = E$ and $(v_i, v_j) \in \mathcal{E}$ indicates that nodes $v_i$ and $v_j$ can communicate with each other. For simplicity, our analysis assumes synchronous updates, the network has no latency, and each node has an identical copy of the data $D$.

---

**Algorithm 3** Distributed Greedy Forward Selection for Two-Layer Networks

---

1: $\mathbf{z}_0^{(j)} := \mathbf{0}, \quad \forall j \in [V]$
2: **for** $k := 1, 2, \ldots$ **do**
3:      # **Step I**: compute a local estimate of the next iterate
4:      **for** $v_j \in \mathcal{V}$ **do**
5:         $\mathbf{q}_k^{(i)} := \underset{\mathbf{q} \in \texttt{Vert}(\mathcal{M}_N^{(i)})}{\arg\min} \ell\left(\frac{1}{k}\left(\mathbf{z}_{k-1} + \mathbf{q}\right)\right)$
6:         $\mathbf{z}_k^{(i)} := \mathbf{z}_{k-1} + \mathbf{q}_k^{(i)}$
7:         **Broadcast:** $L^{(i)} := \ell\left(\mathbf{z}_k^{(i)}\right)$
8:      **end for**
9:      # **Step II**: determine and broadcast the best local iterate
10:     **for** $v_i \in \mathcal{V}$ **do**
11:        $i_k := \underset{i \in [V]}{\arg\min} L^{(i)}$
12:        **if** $i_k = i$ **then**
13:           **Broadcast:** $\mathbf{z}_k := \mathbf{z}_k^{(i)}$
14:        **end if**
15:     **end for**
16:     # **Step III**: update the current, global iterate
17:     **for** $v_i \in \mathcal{V}$ **do**
18:        $\mathbf{u}_k := \frac{1}{k}\mathbf{z}_k$
19:     **end for**
20: **end for**
21: `Stopping Criterion:` $\ell(\mathbf{u}_k) \le \epsilon$

---

Recall that the set of neurons considered by greedy forward selection is given by $\texttt{Vert}(\mathcal{M}_N) = \{\mathbf{\Phi}_i : i \in [N]\}$. In the distributed setting, we assume that the weights associated with each neuron are uniformly and disjointly partitioned across compute sites. More formally, for $j \in [V]$, we define $\mathcal{A}^{(j)}$ as the indices of neurons on $v_j$ and $\mathbf{\Theta}^{(j)} = \{\theta_i : i \in \mathcal{A}^{(j)}\}$ as the neuron weights contained on $v_j$. Going further, we consider $\{\mathbf{\Phi}_i : i \in \mathcal{A}^{(j)}\}$ and denote the convex hull over this subset of neuron activations as $\mathcal{M}_N^{(j)}$. We assume that $\mathcal{A}^{(j)} \bigcap \mathcal{A}^{(k)} = \varnothing$ for $j \neq k$ and that $\bigcup_{j=1}^{V} \mathcal{A}^{(j)} = [N]$.

Algorithm 3 aims to solve the main objective in this work, but in the distributed setting. We maintain a global set of active neurons throughout pruning that is shared across compute nodes, denoted as $\mathcal{S}_k$ at pruning iteration $k$. At each pruning iteration $k$, we perform a local search over the neurons on each $v_j \in \mathcal{V}$, then aggregate the results of these local searches and add a single neuron (i.e., the best option found by any local search) into the global set. Intuitively, Algorithm 3 adopts the same greedy forward selection process from Algorithm 1 but parallelizes it across compute nodes.

**Empirical validation of the distributed implementation.** The centralized and distributed variants of greedy forward selection achieve identical convergence rates with respect to the number of pruning iterations. Despite its impressive empirical results, one of the significant drawbacks of greedy forward selection is that it is slow and computationally expensive compared to heuristic techniques. Distributed greedy forward selection mitigates this problem by parallelizing the pruning process across multiple compute nodes with minimal communication overhead.

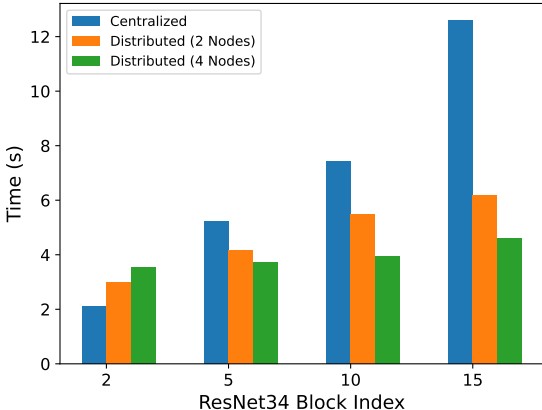

Figure 5: Pruning time for centralized and distributed greedy forward selection applied to different blocks of a ResNet34 architecture on ImageNet.

To practically examine the acceleration provided by distributed greedy forward selection, we prune a ResNet34 architecture (He et al., 2015) on the ImageNet dataset and measure the pruning time for each layer with different greedy forward selection variants. In particular, we select four blocks from the ResNet34 architecture with different spatial and channel dimensions. The time taken to prune each of these blocks is shown in Figure 5. All experiments are run on an internal cluster with two Nvidia RTX 3090 GPUs using the public implementation of greedy forward selection (Ye, 2021).

Distributed greedy forward selection (using either two or four GPUs) significantly accelerates the pruning process for nearly all blocks within the ResNet. Notably, no speedup is observed for the second block because earlier ResNet layers have fewer channels to be considered by greedy forward selection. As the channel dimension increases in later layers, distributed greedy forward selection yields a significant speedup in the pruning process. Given that the convergence guarantees of distributed greedy forward selection are identical to those of the centralized variant, we adopt the distributed algorithm to improve efficiency in most of our large-scale pruning experiments.

