# OpenReview forum: "How Much Pre-training Is Enough to Discover a Good Subnetwork?"
_TMLR — Accepted by TMLR_

### Review · Reviewer_KcNq · 2023-11-29

**Summary Of Contributions:**

The paper considers theoretical bounds on the performance of two-layer fully connected neural networks pruned via greedy forward selection. The paper finds improved bounds on selection error and demonstrates that a minimum amount of pre-training is required for a pruned network to perform well, with this amount depending on dataset size. Experiments on multiple datasets and architectures validate this trend.

**Audience:**

Yes

**Broader Impact Concerns:**

None.

**Claims And Evidence:**

Yes

**Requested Changes:**

**Critical**
* Include a theoretical prediction line in Figure 1


**Would improve**
* Adding error bars or some indication of variability over multiple trials in Figure 1 and the other empirical results
* Further discuss how the results could be extended to other architectures
* Add more experimental results and analysis to the main text from the appendix

**Strengths And Weaknesses:**

**Strengths**

* The paper is generally well-written and easy to understand despite its technical depth.
* Theoretical analysis is comprehensive; most of the paper's technical content is devoted to this. The assumptions made are well justified.
* Coverage of related work is strong.


**Weaknesses**

* The theoretical analysis is limited to two-layer networks, which is understandable. Further discussion on this point would be valuable: how could these results be extended to other architectures?
* It's unclear how the empirical result in Figure 1 corresponds to Theorem 3; it would be good to include a theoretical prediction line in the Figure as well.
* The empirical results are mostly contained in the appendix while the paper mostly focuses on the theoretical results. This is understandable for a theoretical paper; however, I would suggest moving a bit of the theory to the appendix to leave room for a bit more experimental results and analysis in the main text.

---

> ### Author Response · Authors · 2023-12-14
> **Response to Reviewer KcNq**
>
> We thank the reviewer for the thoughtful review. We are encouraged the reviewer finds our theoretical analysis to be comprehensive with well-justified assumptions, and our presentation to be clear and easy to understand. We respond to each of the weaknesses raised by the reviewer below:
>
> 1. We agree that the current theoretical analysis is only applicable to two-layer networks. As the reviewer acknowledged, extending such analysis to deeper networks is non-trivial. That being said, we have added the discussion at the end of the conclusion section about how to extend the analysis to deeper networks, where we stated that a layer-wise analysis would be necessary based on the generalization of the pruning algorithm to deep networks in Ye et al, 2020 cited in our paper. Moreover, we have also included a generalization of the pruning algorithm considered in our paper to convolutional networks in Appendix D.
>
> 2. We thank the reviewer for a great suggestion. In the updated version (Page 11), we have updated Figure 1 with theoretical prediction lines computed based on the condition number of randomly generated Gaussian matrices.
>
> 3. We appreciate that the reviewer recognizes our effort in the theoretical work. As the reviewer suggested, we have moved more experimental results on deep networks to the main text (see Section 6.2).

---

### Review · Reviewer_NNqK · 2023-12-06

**Summary Of Contributions:**

The paper studies the theoretical aspects of pruning in two-layer overparameterized neural networks. It establishes a link between the amount of pre-training needed for this model and the size of the training data. Essentially, larger training datasets necessitate more iterations to ensure that the greedy forward selection algorithm results in a small pruned sub-network with comparable performance. The paper supports these findings with empirical experiments, demonstrating similar patterns in 2-layered binary MNIST trained on image datasets.

**Audience:**

Yes

**Broader Impact Concerns:**

Not relevant for this paper

**Claims And Evidence:**

Yes

**Requested Changes:**

- Extend the empirical section to include the experiments in Appendix E, and elaborate on them to show that in practice bigger datasets require more pre-training outside of 2-layered network scenarios. This point is critical to secure my recommendation for acceptance.

- Include a discussion in Section 4 that addresses the limitations and advantages of the proposed theory within a broader context. This point would strengthen the work in my view, but is not critical to securing a recommendation for acceptance.


minor formatting errors:
- page 23, at the footnote, change Appendix ?? to the real cross-reference
- The title of the paper is broken into 2 lines in a weird place, consider fixing it

**Strengths And Weaknesses:**

Strengths:

- The paper is written clearly and is easy to follow and understand.
- As far as I know, this is the first paper to theoretically show that more pre-training is needed to get a good sub-network when pruning. The topic itself is interesting to the community, and theoretical evidence, even on a simplified model such as two-layered over parameterized networks, is important.

Weaknesses:

- The empirical validation is a weak point of the paper. The theoretical analysis heavily relies on specific assumptions, such as a 2-layer network, a very wide hidden layer, a non-commonly used loss function, and certain assumptions about the data and labels. While these assumptions are standard in the analysis of 2-layered networks, they deviate significantly from the norms in deep networks, raising concerns about the generalizability of the findings. Particularly important is the fact that the paper did not show an iff result: the analysis shows that after many training iterations, all pruning yields good subnetworks. However, it does not show that after fewer training iterations, pruning would result in poor subnetworks -- therefore, in practice, it may be the case that the results do not hold, and extensive empirical validation is required. Such validation should include experimentation with both 2-layered networks and deep neural networks, across various datasets and data sizes, which is crucial for establishing practical applications. Although the main paper features binary MNIST, the inclusion of experiments on CIFAR-10 and ImageNet (which are hinted at in the abstract and discussion but not in the main paper), currently relegated to the appendix, would substantially strengthen the paper and warrant further elaboration on their results.

- The paper would benefit from a discussion of the theory's limitations and advantages within a broader context. For instance, addressing how other factors that might influence the required pre-training are related to or overlooked by the analysis. Potential factors for discussion include the impact of complex data, wider networks requiring more training until convergence, and the diminishing returns effect on data size. This discussion could be seamlessly incorporated at the end of section 4, offering insights into the broader implications of the theory without necessitating additional analysis. I think that many of these points could be easily addressed by the authors (for example, I would think that data complexity could be encoded in the distance of the underlying convex hull), and could strengthen the paper.

---

> ### Author Response · Authors · 2023-12-14
> **Response to Reviewer NNqK**
>
> We thank the reviewer for the thoughtful review. We are glad the reviewer acknowledges the novelty and importance of our work. Below, we address concerns in detail:
>
> 1. We agree that our theory is conservative and only applicable under specific assumptions, and the experimental results were indeed our attempt to show the generalizability of the theoretical findings to various other settings. As the reviewer suggested, we have added the experimental results on deep architectures to the main text, together with the related discussion (see Section 6.2). We believe this inclusion substantially strengthens the paper, as the reviewer noted, and we thank the reviewer for the great suggestion.
>
> 2. We thank the reviewer's suggestion of discussing our result in a broader context. We took the suggestion and added a remark in Section 4 (in teal color), discussing that the main limitation of our work is that the initialization scaling keeps the neural network training in the NTK regime. Moreover, we stated that this scaling is used to control the diameter of the convex hull. Lastly, we connected this assumption with the data complexity, saying that structured data would potentially reduce the diameter of the convex hull and allow the use of scaling beyond the NTK regime.
>
> 3. We thank the reviewer for pointing out these formatting errors. We have fixed them in the revised version.

---

### Review · Reviewer_KEDf · 2023-12-10

**Summary Of Contributions:**

This work proposes a modified greedy forward selection algorithm for pruning and studies its theoretical properties. Specifically, this work provides a theoretical bound of the number of gradient descent pre-training iterations for two-layer fully connected networks with proper activation. It also provides empirical results demonstrating the data size dependency of the pruning performance consistant with the theoretical results.

**Audience:**

Yes

**Broader Impact Concerns:**

N.A.

**Claims And Evidence:**

No

**Requested Changes:**

See Weakness above

**Strengths And Weaknesses:**

Strength:
This work provide new theoretical results regarding the greedy forward selection algorithm.

Weakness:
(i) This work considers a relatively large scale of initialization, i.e.,  \omega_b and \omega_a is fixed as N\to \infty. This initialization leads to an initial loss  of O(\sqrt(N)) dependent on the number of neurons N. However, in Lemma 3 and later results, N is further assumed to be dependent on the initial loss.  In the revision, this inconsistence need to be addressed and the authors should discuss the scale of initialization and its impact on initial loss.

(ii) The theoretical results seems to rely heavily on a NTK-type regime with linear convergence. However, the NTK regime in general generalize poorly. If the theoretical results are only relevant to the NTK-type regime, then it is not so relevant to the training in practice. I suggest the authors to discuss this issue in the revision.

---

> ### Author Response · Authors · 2023-12-14
> **Response to Reviewer KEDf**
>
> We thank the reviewer for the thoughtful review, and we respond to each of the weaknesses raised by the reviewer below:
>
> 1. Thank you for raising this concern; we believe this part was written less clearly in the initial submission. In the statement of Lemma 3 and Theorem 2 in our revised version, we have clarified the requirement on the initialization scale $\omega_a$ and $\omega_b$. In particular, to make Lemma 3 hold, we do require $\omega_b \leq O\left(m^{-\frac{1}{2}}N^{-\frac{3}{4}}\right)$, which was hidden in the proof of Lemma 3 in the previous version. Using this scaling, we can compute that the expected initial loss is constant. In this revised version, we added a more detailed clarification on this point in Section 4.1 in teal color.
>
> 2. We would like to clarify that this scaling is chosen to control the diameter of the convex hull defined over the neural network. In our revised version, we have added a remark in Section 4 in teal color to discuss this issue, and we talked about the potential future work of considering the structure in the training data to relax this restricted initialization scale assumption. From this perspective, we would like to present our paper as a first step toward understanding the relationship between pre-training and pruning. Considering that our paper is the first that studies this direction, we also hope that future works can build upon our results and perform further theoretical investigations into the topic.

---

### Decision · Action_Editor_Dy4u · 2024-01-24

**Recommendation:** Accept with minor revision

**Comment:**

The paper should be accepted, given that it meets the criteria of presenting supported claims with clear evidence. The theoretical contribution add to the understanding of neural network pruning in the studied very limited NTK regime. The empirical study extends this regime to slightly more general case. Although the evidence support claims, the impact and significance of the paper is rather limited due to the overly constrained setting. This limitations prevents me from suggestion certification and journal to conference recommendation.

**Audience:**

The audience interest is very clear. Both theoretical understanding of deep learning and pruning are active research areas. This paper would be very interesting read for anyone working on pruning. Moreover, the insights can be used to design better pruning algorithms.

**Claims And Evidence:**

The manuscript theoretically analyse the greedy forward selection algorithm and specifically present the relationship between the pre-training and the dataset size. All three reviewers find the claims interesting and raised some issues on the evidence. These issues are discussed and clarified during the discussion period. I believe the final decision is clear and the evidence supports the claims. Hence, the paper meets the bar. However, it definitely needs some revision. Specifically,

- Dependence on NTK: It should be discussed more clearly and the limitations should be made prominent.
- Experimental discussion can be extended to further clarify generality beyond the theoretical assumptions. Specifically, an informal discussion on why the results generalise would be interesting.